

# Impact of present and future aircraft NOₓ and aerosol emissions on atmospheric composition and associated direct radiative forcing of climate

Etienne Terrenoire[1,2], Didier A. Hauglustaine[1], Yann Cohen[1], Anne Cozic[1], Richard Valorso[3], Franck Lefèvre[4], and Sigrun Matthes[5]

[1]Laboratoire des Sciences du Climat et de l'Environnement (LSCE), UMR 8212, Gif-sur-Yvette, France.
[2]Now at Office National d'Etudes et de Recherches Aérospatiales (ONERA), DMPE, Université Paris-Saclay, Palaiseau, France.
[3]Univ. Paris-Est-Créteil and Université de Paris, CNRS, LISA, F-94010 Créteil, France.
[4]Laboratoire Atmosphères, Milieux, Observations Spatiales (LATMOS), UMR 8190, Paris, France.
[5]Deutsches Zentrum für Luft-und Raumfahrt e.V., DLR Institut für Physik der Atmosphäre, Oberpfaffenhofen, 82334 Wessling, Germany

*Correspondence to*: D. A. Hauglustaine (didier.hauglustaine@lsce.ipsl.fr)

**Abstract.**

Aviation NOₓ emissions have not only an impact on global climate by changing ozone and methane levels but also contribute to deteriorate local air quality. In order to properly assess the co-benefit with air quality improvement and the trade-off with the climate change associated with $CO_2$, it appears essential to better quantify the climate impact of aircraft NOₓ emissions. A new version of the LMDZ-INCA global model, including both tropospheric and stratospheric chemistry and the sulfate-nitrate-ammonium cycle, is applied to re-evaluate the impact of aircraft NOₓ and aerosol emissions on climate. The results confirm that the efficiency of NOₓ to produce ozone is very much dependent on the injection height. For the baseline simulation and reference scenario this efficiency is 6.6 $TgO_3/TgN$. The efficiency increases with the background methane and NOₓ concentrations and with decreasing aircraft NOₓ emissions. The same findings translate to the associated ozone radiative forcing which exhibits a fairly constant value per ozone mass change of 3.4 $mW/m^2/TgO_3$. The methane lifetime variation is less sensitive to the aircraft NOₓ emission location than the ozone change. The change in $CH_4$ mixing ratio itself represents 75% of the total methane forcing. The indirect changes through long-term tropospheric ozone, stratospheric water vapour and methane oxidation to $CO_2$ contribute for 19%, 4% and 1%, respectively. The net NOₓ radiative forcing ($O_3 + CH_4$) is largely affected by the revised $CH_4$ radiative forcing formula which increases the total $CH_4$ negative forcing by 15%. As a consequence, the ozone positive forcing and the methane negative forcing largely offset each other resulting in a slightly positive forcing for the present-day. However, in the future, the net forcing turns to negative due essentially to higher methane background concentrations. Additional radiative forcings involving particle formation arise from aircraft NOₓ emissions since the increased OH concentrations are responsible for an enhanced conversion of $SO_2$ to sulfate particles. Aircraft NOₓ emissions also increase the formation of nitrate particles in the lower troposphere. However, in the upper-troposphere, increased sulfate concentrations favor the titration of ammonia leading to lower ammonium nitrate concentrations. The total forcing from sulfate and nitrate aerosols associated with NOₓ emissions is negative and is estimated to -3.0 $mW/m^2/TgN$ for the present-day. When these aerosol radiative forcings are considered, the total NOₓ forcing turns from a positive value to a negative value even for present-day conditions. Hence, total radiative forcing from aircraft emissions associated with changes in atmospheric chemistry and direct aerosols forcings is negative for both present-day and future (2050) conditions. NOₓ emissions only cause a negative forcing representing about 45% of the total forcing. The negative forcing associated with sulfates largely dominates the effect of the other particles. The sulfate direct radiative forcing is estimated to be associated for about 50% with the direct $SO_2$ and $SO_4$ aircraft emissions and for about 50% with the increased conversion of $SO_2$ to $SO_4$ at higher OH concentrations, and hence related to the NOₓ emissions. The net effect of decreasing (resp. increasing) the flight altitude by 2000 ft (about 610 m) is to increase (resp. decrease) the total negative forcing by 57% (resp. 65%). The variation of the total forcing with flight altitude is dominated by the high sensitivity of the ozone positive forcing to the altitude of the perturbation. Several mitigation options involving flight aircraft operation and cruise altitude changes, traffic growth, engine technology, and fuel type, exist to reduce the climate impact of aircraft NOₓ emissions. However, the climate forcing of aircraft NOₓ emissions is likely to be small or even switch to negative (cooling) depending on atmospheric NOₓ or $CH_4$ future background concentrations or when the NOₓ impact on sulfate and nitrate particles is considered. There remain large uncertainties on the NOₓ net impact on climate effect calculation. Nevertheless, the results suggest that reducing aircraft NOₓ emissions is primarily beneficial for improving air quality. For climate consideration, one option to reduce uncertainties in mitigation strategies might be to prioritize the reduction of $CO_2$ aircraft emissions which have a well-established and long-term impact on climate, however this reduces the overall mitigation potential.



## 1 Introduction

Air traffic emissions represent a sizeable contribution to global anthropogenic climate change (Lee et al., 2021) and also to regional surface air pollution, in particular around airports (Yim et al., 2015). Aircraft release in the atmosphere not only gaseous compounds such as carbon dioxide ($CO_2$), nitrogen oxides ($NO_x$), hydrocarbons (HC), sulfur dioxide ($SO_2$) and water vapour ($H_2O$), but also particulate material composed of ice crystals, soot particles (Black Carbon, BC) and sulfates ($SO_4$) (e.g., Kärcher, 2018; Lee et al., 2021). There is a wide range of
spatial and temporal scales associated with atmospheric perturbations due to aircraft emissions. It ranges from the local and plume scales for chemical species, aerosols, and contrail-cirrus formation to the global scale for methane and carbon dioxide perturbations; and from a few minutes after emission up to several decades (Brasseur et al., 2016). Evaluating the global chemical and climate perturbations associated with aircraft emissions therefore appears as a complex issue.

Due to this range of scales and also to the different nature of the perturbations of the climate system involved, a distinction is usually made between the $CO_2$ and the non-$CO_2$ climate impacts of aviation. These climate forcings were recently reassessed by Lee et al. (2021). For 2018, the net aviation Effective Radiative Forcing (ERF) of historic aviation emissions until 2018 was estimated in this recent study to 101 mW/m² with major contributions
from contrail cirrus (57.4 mW/m² or 57% of the total forcing), $CO_2$ (34.3 mW/m², 34%), and $NO_x$ (17.5 mW/m², 17%). Non-$CO_2$ terms represent a net positive forcing that accounts for more than half (66%) of the aviation total radiative forcing of climate. In contrast to the $CO_2$ forcing which is relatively well determined except for some methodological issues (Terrenoire et al., 2019; Boucher et al., 2021), non-$CO_2$ forcing terms contribute about 8 times more than $CO_2$ to the uncertainty in the aviation total forcing (Lee et al., 2021).

From these non-$CO_2$ climate forcings associated with aircraft emissions, nitrogen oxides ($NO_x$) play a particular role. Indeed, not only do they affect climate by changing the atmospheric concentration of ozone ($O_3$) and methane ($CH_4$) (e.g., Hauglustaine et al., 1994; Brasseur et al., 1998; Holmes et al; 2011; Myhre et al., 2011), two important greenhouse gases, but they also have an impact on local air quality and human health, both through emissions in
and around airports and from emissions at higher altitude affecting near-ground background concentrations (e.g., Barrett et al., 2010; Hauglustaine and Koffi, 2012; Cameron et al., 2017). For this reason, it has been assumed that emissions standards for $NO_x$ emissions set since the 1980s by the International Civil Aviation Organization (ICAO) were not only protecting local air quality but also had co-benefits for climate change mitigation (Skowron et al., 2021). A technological difficulty faced by aircraft industry today is that reducing $NO_x$ emissions tends to
increase fuel burn, hence resulting in increased $CO_2$ emissions and a climate penalty. There is however a possibility that technological development could lead simultaneously to $CO_2$ and $NO_x$ emissions reduction. Hence, a better quantification of the aircraft $NO_x$ effect on climate is needed to determinate if the climate effect of $CO_2$ increase linked to new technology could be compensated by the associated $NO_x$ reduction.

Emissions of $NO_x$ into the troposphere result in a short-term increased photochemical ozone production (resulting in a positive climate forcing or warming), and a long-term increased oxidation of atmospheric methane through reaction with the hydroxyl radical (OH) (resulting in a negative climate forcing or cooling) (Fuglestvedt et al., 1999; Naik et al., 2022). In addition, the aforementioned methane reduction results in a long-term reduction in tropospheric $O_3$ (cooling) and a long-term reduction in $H_2O$ in the stratosphere (cooling). In the lower troposphere,
the net effect of $NO_x$ emissions is dominated by the increased methane destruction and a negative forcing is predicted. In the upper-troposphere, at aircraft cruise altitudes, the ozone production is about 5 times more efficient per molecule of $NO_x$ emitted (e.g., Hauglustaine et al., 1994; Derwent et al., 2001; Hoor et al., 2009; Dahlmann et al., 2011) and a positive net radiative forcing of climate is generally associated with aircraft $NO_x$ emissions (e.g., Holmes et al., 2011; Myhre et al., 2011; Hoor et al., 2009; Søvde et al., 2014; Brasseur et al., 2016; Lee et al.,
2021). Based on an in-depth literature assessment, Lee et al. (2021) provided the most recent estimate of these forcings and derived a net $NO_x$ radiative forcing from all emissions released by aviation of 17.5 (0.6, 28.5) mW/m² in 2018, decomposed into a positive short-term tropospheric ozone forcing of 49.3 (32, 76) mW/m² and a negative methane forcing of -34.9 (-65, -25.5) mW/m². The aircraft $NO_x$ net radiative forcing of climate is positive but this net effect is the sum of two forcings of opposite sign, distinct geographic distributions and each of them associated
with large uncertainties.

Recently, Skowron et al. (2021) reinvestigated the aircraft $NO_x$ climate forcing based on various future scenarios for both surface and aircraft $NO_x$ emissions using the MOZART-3 global chemistry-climate model (Kinnison et al., 2007). They found that in all their future (2050) simulations and even for "present-day" (2006) simulations
under certain conditions, the net radiative forcing from aircraft $NO_x$ could turn negative. This finding is essentially associated with the revised expression for methane direct radiative forcing from Etminan et al. (2016) which increases, for instance, the methane forcing by 24.5 % for a halving of its atmospheric concentration compared to previous formulationss. However, another major uncertainty associated with the impact of $NO_x$ emissions on atmospheric composition arises from the non-linear character of the tropospheric chemistry. This feature makes
the impact of aircraft $NO_x$ dependent on the background atmospheric concentrations and hence sensitive to anthropogenic surface emissions of $NO_x$, CO and hydrocarbons or even to natural emissions such as lightning $NO_x$ (Holmes et al., 2011; Skowron et al., 2021). It also makes the impact of aircraft $NO_x$ very dependent on the location of the emission and on the season (Stevenson et al., 2004; Stevenson and Derwent, 2009; Gilmore et al., 2013;





Skowron et al., 2013; Søvde et al., 2014). Since the ozone production sensitivity differs from the methane destruction sensitivity, the positive short-term ozone forcing associated with aircraft $NO_x$ can be overwhelmed by the methane negative forcing, providing a negative net radiative forcing of climate (Stevenson and Derwent, 2009; Skowron et al., 2021).

Other effects of aircraft $NO_x$ emissions, less accounted for in earlier studies, include the role played by tropospheric aerosols and in particular the impact on secondary inorganic aerosols such as nitrates and sulfates (Unger, 2011; Pitari et al., 2015; Brasseur et al., 2016). Increased $NO_x$ emissions from aircraft have indeed the potential to form ammonium nitrates particles. However, the changes in oxidants and the direct emission of $SO_2$ by aircraft also increase the formation of sulfate particles with possible implications on nitrate concentrations (Unger et al., 2013; Righi et al., 2013; 2016). These indirect forcings associated with aircraft $NO_x$ emissions are however complex and 135 need further investigation since they can provide additional negative direct forcings of climate. In order to account for the role played by secondary inorganic aerosols and other interactions involving gas-phase and aerosols chemistry, the global models used to asses the impact of aircraft $NO_x$ emissions need to include both gas phase chemistry, tropospheric aerosols and in particular the role played by the sulfate-nitrate-ammonium cycle.

The aim of the present study is to provide a comprehensive and updated model-based analysis of the impact of aircraft $NO_x$ emissions on atmospheric composition and associated radiative forcing of climate. The LMDZ-INCA global model, including both tropospheric and stratospheric chemistry as well as tropospheric aerosols, is applied. An earlier and less mature version of this model has been precedently used by Koffi et al. (2010) and Hauglustaine and Koffi (2012), or in several model intercomparisons (Hoor et al., 2009; Hodnebrog et al., 2011; 2012; Myhre 145 et al., 2011) in order to investigate the impact of $NO_x$ transport emissions on atmospheric composition and climate. This earlier version of the model only included tropospheric gas phase chemistry and used a coarser vertical resolution. The new version of LMDZ-INCA used in this study allows to revisit the impact of aircraft $NO_x$ on ozone production, changes in oxidants and methane destruction but also the impact on the secondary inorganic aerosol distributions. These earlier studies will provide a point of comparison for this new assessment. This study 150 focuses on the effect of aircraft $NO_x$ emissions but also includes aircraft $SO_2$ and aerosols emissions in order to account for the effect of the sulfate-nitrate-ammonium cycle and consider the potential effect linked to heterogeneous chemistry. Similarly, the direct water vapour aircraft emission is also included in order to account for the role of stratospheric water vapour on atmospheric oxidants and in link also with the methane indirect effect on $H_2O$.

Moreover, another aim of this paper is to investigate the sensivivity of the aircraft $NO_x$ net radiative forcing of climate to various mitigation options. Motivated by earlier work (e.g., Unger, 2011; Hodnebrog et al., 2012; Matthes et al., 2021; Skowron et al., 2021) we assess in particular the sensivity of the $NO_x$ forcing to a variation of the aircraft emission injection height and to background (present versus future) atmospheric concentrations. In 160 the future, several scenarios are considered in order to investigate low and high assumptions in air-traffic growth and fuel burn efficiencies. Other scenarios will also illustrate more specifically the impact of desulfurized jet fuel as a mitigation option or engines with ultra-low $NO_x$ combustor technology. For all these "present-day" and future (2050) scenarios, the changes in atmospheric composition are illustrated and the radiative forcings of climate associated with $NO_x$ emissions ($O_3$ and $CH_4$ direct and indirect forcings) and with aerosol direct effects are 165 calculated.

The remainder of this paper is organized as follows. In **Section 2**, we present the aircraft emission inventories prepared and introduced in the global chemistry-climate model for both the "present-day" baseline simulation and for the future (2050) baseline and mitigation scenarios. In **Section 3**, we provide a description of the LMDZ-INCA 170 chemistry-climate model used in this study along with a description of the radiative forcing calculations and modelling set-up. In addition, **in Section 3**, we also summarize the model performance comparing the model results with ozone soundings and with the IAGOS (In-service Aircraft for a Global Observing System) measurements of ozone and carbon monoxide concentrations in the upper-troposphere and lower-stratosphere. **In Section 4**, we present the atmospheric composition perturbations associated with the "present-day" aircraft 175 emissions and in **Section 5** the perturbations associated with future aircraft emissions under different scenarios. We then present the radiative forcings of climate associated with the changes in atmospheric composition in **Section 6**. Finally, in **Section 7**, we discuss conclusions drawn from this study.

## 2 Aircraft emissions

180

The global three-dimensional and time varying aircraft emission inventories used in this study are essentially based on the previous Reducing Emissions from Aviation by Changing Trajectories for the benefit of Climate (REACT4C) (Matthes et al., 2012) and Quantifying the Climate Impact of Global and European Transport Systems (QUANTIFY) (Lee et al., 2010) European Union (EU) projects. These inventories are based on the fuel-flow 185 model PIANO (Project Interactive Analysis and Optimization model) and the global emission model FAST (Future Aviation Scenario Tool) with air traffic movements coming from radar data for flights for Europe and North America and the the Official Airline Guide (OAG) database for the remaining global flight movements (Lee et al., 2009; Owen et al., 2010). For this specific study, the methodology used to derive emissions for the global



chemistry-transport model LMDZ-INCA is futher described in the following Section 2.1 for "present-day" baseline emissions and in Section 2.2 for the future (2050) emission scenarios.

## 2.1 "Present-day" baseline emissions

The aircraft emission inventory used for "present-day" baseline emissions is based on the EU project REACT4C (Matthes et al., 2012, 2021; Søvde et al., 2014; Grewe et al., 2014) and is representative of the year 2006. This inventory will be refered to as REACT4C_2006 in this paper. The REACT4C_2006 data includes three-dimensional gridded distributions of travelled-km, fuel consumption, as well as $CO_2$, $NO_2$, and BC (Black Carbon, soot) emissions. Flight data are derived from CAEP8 data using the "great circle distance" method corrected with the CAEP8 formula. This dataset is available on a latitude-longitude-altitude grid for 12 months with a horizontal 200 resolution of 1°x1° and a 610 m vertical resolution. In this inventory, the global mean $NO_x$ Emission Index (EI) is 12.1 $gNO_2$/kg fuel (12.7 $gNO_2$/kg below 1800 m and 12.3 $gNO_2$/kg above 8400 m). For BC, the mean EI is 0.023 $gBC$/kg fuel (0.046 $gBC$/kg below 1800 m and 0.015 $gBC$/kg above 8400 m). Additional species are needed to run the global chemistry-transport model. For $H_2O$, carbon monoxide (CO) and total non-methane hydrocarbons (HC), we use the three-dimensional emissions from the AERO2K project inventory (Eyers et al, 2005). The EIs 205 for these species are derived for the AERO2K vertical levels (500 feet vertical resolution) and interpolated onto the REACT4C vertical levels. Based on the REACT4C fuel burn we then derive the $H_2O$, CO and HC three-dimensional and monthly emissions representative of the year 2006. For HC we use the speciation given by FAA (2009) in order to derive the emissions of the LMDZ-INCA model individual hydrocarbons. For Organic Carbon (OC), $SO_2$ and $SO_4$, we use the mean EIs reported by Lee et al. (2010). For OC, Lee et al. (2010) provide a range 210 of 0.0065-0.05 g/kg. As was done by Balkanski et al. (2010) and Righi et al. (2016), we choose the highest EI value of 0.05 g/kg fuel and determine a maximum value for organic carbon produced from aircraft. For $SO_2$ and $SO_4$, the mean EIs are respectively 0.8 and 0.04 g/kg fuel. From REACT4C_2006, we derive global EIs of 3.15 $kgCO_2$/kg fuel, 1.23 $kgH_2O$/kg fuel, 3.25 $gCO$/kg fuel, and 0.405 $gHC$/kg fuel, which compares well to the $EI_{CO2}$ and $EI_{H2O}$ used in Lee et al. (2021) recent review. They are somewhat lower than the EIs used in Lee et al. (2021) 215 for BC ($EI_{BC}$=0.03 $gBC$/kg fuel), $NO_x$ ($EI_{NOx}$=14.12 $gNO_2$/kg fuel) and $SO_2$ ($EI_{SO2}$=1.2 g/kgfuel). **Table 1** summarizes the total emissions for the baseline REACT4C_2006 inventory.

The REACT4C_2006 inventory is chosen for the "present-day" baseline simulation since this inventory provides, in addition to the baseline emissions, two additional mitigation inventories. These motigation scenarios based on 220 the original REACT4C_2006 emissions were used in this study in order to investigate the sensitivity of the chemical perturbations and associated radiative forcings to a cruise altitude variation (Søvde et al., 2014, Matthes et al., 2021). REACT4C_PLUS corresponds to the original REACT4C_2006 inventory with flight altitude increased by 2000 feet (610 m) while in REACT4C_MINUS the flight altitudes been decreased by 2000 feet. While the total distance flown is approximately equal in all REACT4C inventories, as fuel efficiency increases 225 with altitude, the fuel use is 178, 177 and 181 Tg/yr in the baseline, REACT4C_PLUS and REACT4C_MINUS inventories, respectively. As a consequence, the total $NO_x$ emissions are respectively 0.71, 0.72 and 0.71 TgN/yr.

In order to compare the future perturbations based on the QUANTIFY project emissions (see next section) to the "present-day" perturbations, another reference inventory has been used in this paper. This inventory labelled 230 QUANTIFY_2000 is based on the QUANTIFY assumptions as described in Owen et al. (2010). In particular, this QUANTIFY_2000 reference is scaled based on IEA sales justifying the higher fuel burn compared to REACT4C_2006. The QUANTIFY emission inventory has been extended to the additional species needed in the global chemistry-transport model as described above. As a consequence of the 20% higher fuel consumption and different assumptions for EIs (for BC and $NO_x$) in this inventory, the total emissions of primary species are higher 235 by 18-25% than those provided for REACT4C_2006 (**Table 1**). This inventory is only used for the sake of comparison with the more recent REACT4C_2006 inventory and as baseline for the future perturbations for which 2050 QUANTIFY inventories are used.

**Table S1** gives the total global emissions for the REACT4C_2006 and QUANTIFY_2000 inventories used in this 240 study and compares to the Aviation Climate Change Research Initiative (ACCRI) 2006 Aviation Environmental Tool (AEDT) (Brasseur et al., 2016) and Community Emissions Data System (CEDS) 2006 (Hoesly et al., 2018) inventories. The AEDT and CEDS inventories use the US governmental's Volpe National Transportation Systems Centre data. It should be noted that these inventories not only differ in terms of total fuel use and global emissions but also in terms of vertical distributions as described in Skowron et al. (2013).


## 2.2 Future 2050 emission scenarios

In this study, the future aircraft emission inventories prepared during the QUANTIFY EU project, and representative of the year 2050 are used (Owen et al., 2010). These scenarios were developed more than 10 years 250 ago based on earlier economic assumptions regarding future Gross Domestic Product (GDP) growth and aviation demands in various regions according to the former IPCC Special Report on Emission Scenarios (SRES) storylines (Nakicenovic and Swart, 2000). These scenarios are still used in this study since they can be considered as benchmark scenarios, used in numerous former model simulations and intercomparisons (e.g., Koffi et al., 2010; Hodnebrog et al., 2011; 2012; Righi et al., 2016). These scenarios compare fairly well in terms of total future



aircraft emissions to the more recent inventories. Three different aircraft emission inventories are selected in this study: the A1B (labelled QUANTIFY_A1 in the following), B1 (QUANTIFY_B1) and B1_ACARE (QUANTIFY_B1_ACARE) scenarios. These scenarios are described in details in Owen et al. (2010) and are only briefly summarized in the following. The scenario A1B is representative of an intense growth of the aviation sector during the first part of the century where global demand is driven by growth of the global economy. Due to

technological improvements and introduction of new and less polluting airplanes within the overall fleet, the fuel efficiency improvement is assumed to be approximately 1% yr$^{-1}$ over the period 2000-2050. In contrast, the scenario B1 is a mitigation scenario in which the propensity to travel is reduced due to a goal to limit the environmental impact of aviation and to improve local air quality in particular. The fuel efficiency improves by 1% yr$^{-1}$ over the 2000-2020 period and increases by 1.3 % yr$^{-1}$ after 2020. This leads to a significant reduction of

$NO_x$, $SO_2$ and BC global emissions in 2050 for this scenario. Finally, assuming the technology targets of the Advisory Council for Aeronautical Research in Europe (ACARE) in 2010, the alternative scenario B1_ACARE is an ambitious mitigation scenario which main goal is to limit the environmental impact of aviation. In this scenario the fuel efficiency assumptions are further tightened and the efficiency improves by 2.1% yr$^{-1}$ after 2020. As a result, the fuel consumption is divided by more than a factor 2 compared to the A1B scenario in 2050. The

reduction hypotheses behind this extreme goal are the ACARE 2050 goals, which for example aim at a 40% improvement in aircraft fuel efficiency (with a further 10% improvement from air traffic management) compared to an equivalent new aircraft introduced in 2000. The variables available from the QUANTIFY three-dimensional inventories are the fuel burn, $NO_x$ and BC emissions. The highest $NO_x$ EIs derived from these inventories are for the A1B scenario and reaches 15.2 $gNO_2$/kg fuel (20 $gNO_2$/kg below 1800 m and 13.3 $gNO_2$/kg above 8400 m).

For the mitigation scenarios B1 and B1_ACARE, the technology improvement strongly impacts the $NO_x$ emissions and the EIs decrease by about a factor of 2 (i.e., 7.9 $gNO_2$/kg fuel and 7.3 $gNO_2$/kg fuel for B1 and B1_ACARE, respectively). For BC, the EI remains fairly constant in the various QUANTIFY future scenarios (0.022, 0.021 and 0.019 gBC/kg fuel for A1B, B1 and B1_ACARE, respectively). The EIs for other species (i.e. $H_2O$, CO, HC, OC, $SO_2$, $SO_4$) are assumed to be similar to those used for the REACT4C emission inventory.


         In addition to the QUANTIFY A1, B1 and B1_ACARE scenarios, two other mitigation scenarios are used for the perturbation simulations. The QUANTIFY_LowNOx scenario is identical to the A1 scenario with the $NO_2$ emission index divided by a factor of 2. This scenario is intended to illustrate one of the ACARE objectives of reducing $NO_x$ emissions by the 2050 time-horizon compared to 2000. Similarly, the A1_Desulfurized scenario

corresponds to the QUANTIFY_A1 scenario, with $SO_2$ and sulfate emissions imposed to 0 to illustrate the impact of desulfurized fuels on the chemical composition perturbations and climate. **Table 2** summarizes the total global emissions in 2050 for the different considered species and for the QUANTIFY_A1, QUANTIFY_B1 and QUANTIFY_B1_ACARE scenarios. As expected, for the QUANTIFY_A1 scenario, an increase in overall fuel consumption compared to the QUANTIFY_2000 of a factor of 3 is obtained and a factor 4 for NOx. For the

QUANTIFY_B1 and QUANTIFY_B1_ACARE scenarios the $NO_x$ emissions are reduced by a factor of 3.3 and 4.8, respectively, compared to the A1B scenario. The BC emissions are reduced by a factor of 1.8 and 2.5, respectively, and the $SO_2$ emissions by a factor of 1.7 and 2.3 respectively.

         The total emissions from the QUANTIFY inventories are compared for 2050 to the recent Shared Socioeconomic

Pathways (SSP) scenarios (Hoesly et al., 2018) and to the ACCRI AEDT scenarios (Brasseur et al., 2016) at **Table S2**. The ACCRI 2050-Base scenario is a high emission scenario providing emissions even higher than the QUANTIFY_A1 scenario, in particular in terms of BC emissions (81% higher emissions), $SO_2$ (87% higher) and to a lesser extent for $NO_x$ (20% higher). We also note for comparison that Skowron et al. (2021) recently assumed a high air-traffic growth and low technology development reaching a high $NO_x$ emission of 5.59 TgN yr$^{-1}$ in 2050

in their "high scenario" and a low air-traffic growth and optimistic technology development reaching 2.17 TgN yr$^{-1}$ in 2050 for the "low scenario", intermediate between the A1B and B1 scenarios in terms of $NO_x$ emissions. The ACCRI 2050-S1 scenario also provides emissions intermediate between the A1B and B1 scenarios in terms of global $NO_x$ and BC emissions. The SSP3-7.0 scenario, used as a reference in the AerChemMIP model intercomparison (Collins et al., 2017), also provides $NO_x$ emissions intermediate between the QUANTIFY_A1

and QUANTIFY_B1 and BC and $SO_2$ emissions close to QUANTIFY_A1. The mitigation scenario SSP1-2.6, also used as a mitigation option for the AerChemMIP simulations, is close to the QUANTIFY_B1_ACARE in terms of global emissions. This comparison suggests that the benchmark QUANTIFY scenarios used in this study are generally consistent with more recent aircraft scenarios in terms of global mean emissions and provide a reasonable estimate for both baseline and mitigation scenarios.


**3 The LMDZ-INCA model**

**3.1 Model description**

The LMDZ-INCA global chemistry-aerosol-climate model couples on-line the LMDZ (Laboratoire de Météorologie Dynamique, version 6) General Circulation Model (Hourdin et al., 2020) and the INCA (INteraction with Chemistry and Aerosols, version 5) model (Hauglustaine et al., 2004; 2014). The interaction between the atmosphere and the land surface is ensured through the coupling of LMDZ with the ORCHIDEE (ORganizing Carbon and Hydrology In Dynamic Ecosystems, version 1.9) dynamical vegetation model (Krinner et al., 2005).

In the present configuration, we use the "Standard Physics" parameterization of the GCM (Boucher et al., 2020).



The model includes 39 hybrid vertical levels extending up to 70 km. The horizontal resolution is 1.9° in latitude and 3.75° in longitude. The primitive equations in the GCM are solved with a 3 min time-step, large-scale transport of tracers is carried out every 15 min, and physical and chemical processes are calculated at a 30 min time interval. For a more detailed description and an extended evaluation of the GCM we refer to Hourdin et al. (2020). The large-scale advection of tracers is calculated based on a monotonic finite-volume second-order scheme (Van Leer, 1977; Hourdin and Armengaud 1999). Deep convection is parameterized according to the scheme of Emanuel (1991). The turbulent mixing in the planetary boundary layer is based on a local second-order closure formalism. The transport and mixing of tracers in the LMDZ GCM have been investigated and evaluated against observations for both inert tracers and radioactive tracers (e.g., Hourdin and Issartel, 2000; Hauglustaine et al., 2004) and in the framework of inverse modelling studies (e.g., Bousquet et al., 2010; Zhao et al., 2019).

INCA initially included a state-of-the-art $CH_4$-$NO_x$-CO-NMHC-$O_3$ tropospheric photochemistry (Hauglustaine et al., 2004; Folberth et al., 2006). The tropospheric photochemistry and aerosols scheme used in this model version is described through a total of 123 tracers including 22 tracers to represent aerosols. The model includes 234 homogeneous chemical reactions, 43 photolytic reactions and 30 heterogeneous reactions. Please refer to Hauglustaine et al. (2004) and Folberth et al. (2006) for the list of reactions included in the tropospheric chemistry scheme. The gas-phase version of the model has been extensively compared to observations in the lower troposphere and in the upper troposphere. For aerosols, the INCA model simulates the distribution of aerosols with anthropogenic sources such as sulfates, nitrates, black carbon (BC), organic carbon (OC), as well as natural aerosols such as sea salt and dust. The heterogeneous reactions on both natural and anthropogenic tropospheric aerosols are included in the model (Bauer et al., 2004; Hauglustaine et al., 2004; 2014). The aerosol model keeps track of both the number and the mass of aerosols using a modal approach to treat the size distribution, which is described by a superposition of 5 log-normal modes (Schulz, 2007), each with a fixed spread. To treat the optically relevant aerosol size diversity, particle modes exist for three ranges: sub-micronic (diameter $< 1\mu m$) corresponding to the accumulation mode, micronic (diameter between 1 and $10\mu m$) corresponding to coarse particles, and super-micronic or super coarse particles (diameter $> 10\mu m$). This treatment in modes is computationally much more efficient compared to a bin-scheme (Schulz et al., 1998). Furthermore, to account for the diversity in chemical composition, hygroscopicity, and mixing state, we distinguish between soluble and insoluble modes. In both sub-micron and micron size, soluble and insoluble aerosols are treated separately. Sea-salt, $SO_4$, $NO_3$, and methane sulfonic acid (MSA) are treated as soluble components of the aerosol, dust is treated as insoluble, whereas black carbon (BC) and organic carbon (OC) appear both in the soluble and insoluble fractions. The ageing of primary insoluble carbonaceous particles transfers insoluble aerosol number and mass to soluble with a half-life of 1.1 day. Ammonia and nitrates aerosols are considered as described by Hauglustaine et al. (2014). The aerosol component of the LMDZ-INCA model has been extensively evaluated during the various phases of AEROCOM (e.g., Gliß et al., 2021; Bian et al., 2017).

Earlier versions of the LMDZ-INCA model only including gas-phase tropospheric chemistry have been previously used to assess the impact of subsonic aircraft on tropospheric ozone (Koffi et al., 2010; Hauglustaine and Koffi, 2012). These previous versions of the LMDZ-INCA model were prescribing the ozone distribution to satellite observations above a potential temperature of 380K, providing a strong constraint on the ozone perturbation at aircraft flight altitudes. In order to assess the impact of aircraft emissions on atmospheric composition, the model has been extended to include an interactive chemistry in the stratosphere and mesosphere. Chemical species and reactions specific to the middle atmosphere have been included. A total of 31 species were added to the standard chemical scheme, mostly belonging to the chlorine and bromine chemistry, and 66 gas phase reactions and 26 photolytic reactions. Water vapour is now affected by both physical processes in LMDZ and, in the stratosphere, an additional $H_2O$ tracer is introduced in INCA in order to account for photochemical production and destruction. In addition, heterogeneous processes on Polar Stratospheric Clouds (PSCs) and stratospheric aerosols are parameterized in INCA following the scheme implemented in Lefèvre et al. (1994). PSCs are first predicted as a function of $H_2O$ and $HNO_3$ local partial pressures, using the saturation vapour pressures for type I PSC (nitric acid trihydrate crystals) and for type II water-ice PSC (Hanson et al. 1988, Carslaw et al. 1995). The excess of $H_2O$ and $HNO_3$ is removed from the gas phase when saturation occurs and is used to compute the surface area concentration in the PSC region. Heterogeneous reaction rates are calculated explicitly, as a function of the surface area available, mean molecular velocity, and the reaction probabilities. Furthermore, the PSC scheme includes sedimentation of the cloud material. The fallout of PSC particles affects the vertical distribution of $H_2O$, $HNO_3$, and HCl. Condensed species are returned to the gas phase when clouds evapourate. In the presence of PSCs, the heterogeneous reactions convert bromine and chlorine reservoirs (HCl, HBr, $ClONO_2$, $BrONO_2$) into reactive species ($Cl_2$, $ClNO_2$, HOCl, $Br_2$, $BrNO_2$, HOBr) based on 9 additional heterogeneous reactions introduced in the chemical scheme. The distribution of stratospheric aerosols is prescribed according to the CCMI exercise (Thomason et al., 2018).

## 3.2 Model evaluation

We refer to previous publications for a general evaluation of the LMDZ-INCA model performances and comparison against observations for both gas phase chemistry and aerosols (e.g., Brunner et al., 2003, 2005; Hauglustaine et al., 2004; Folberth et al., 2006; Myhre et al., 2013; Hauglustaine et al., 2014; Koffi et al. 2016). The version of the LMDZ-INCA model used in this study and extended to the stratosphere has been evaluated by comparing model outputs with observations, in particular in the upper-troposphere and lower-stratosphere, before





the model has been applied to investigate the impact of aircraft emissions. In this study, we summarize this evaluation for two key observational datasets.

**Figure 1** synthetizes the assessment of the modelled ozone seasonal cycles against ozonesondes as compiled by Tilmes et al. (2012) and with respect to the latitude and pressure domains. We present Taylor diagrams to summarize the model biases and correlations with the dataset. The model results are interpolated on both the horizontal and vertical to mimic the 42 ozonesonde stations observations over the 1995-2011 observational period. In matter of yearly mean biases, we distinguish two distribution modes for the three geographical domains,
although it is less visible in the tropics: the $100 - 200$ hPa domain (corresponding to the lower stratosphere in the middle and high latitudes), and the other pressure intervals throughout the troposhere. On one hand, the $200 - 1000$ hPa mean values show a tendency towards negative biases, especially at high-latitudes where the modelled yearly averages spread from about 60% to 100% with respect to the observations, contrasting with the 80 to 110 % interval elsewhere. On the other hand, most of $100 - 200$ hPa yearly mean values are positively biased. In the
middle and high latitudes, almost all the significant positive biases concern this pressure domain.

In matter of correlation between observed and modelled seasonal cycles, Pearson's $r$ coefficient mainly spreads from 0.60 up to 0.95, and is mostly greater than 0.8. Outside the tropics, this metric also shows distinct distribution modes between the same pressure domains. Higher correlations are reported in the $100 - 200$ hPa interval where
most $r$ values are greater than 0.9 and part of them even reach the 0.99 value. Note that the six high-latitude points showing a poor correlation and a strong positive bias correspond to the only three stations located in the southern polar region (Marambio, 64°S; Syowa, 69°S; Neumayer, 70°S). Consequently, although the simulation tends to overestimate lower-stratospheric ozone in the middle and high latitudes, it reproduces the seasonality particularly well outside the southern pole.

**Figure 2** provides, again as Taylor diagrams, the assessment of the modelled geographical distribution against IAGOS observations for ozone and CO, averaged over the periods December 1994 – November 2017 and December 2001 – November 2017 respectively (Cohen et al., 2018). The IAGOS (In-service Aircraft for a Global Observing System, www.iagos.org) is a European research infrastructure performing *in situ* measurements on
board several passenger aircraft (Petzold et al., 2015). Amongst the observed variables, ozone and CO have been monitored so far since August 1994 and December 2001, respectively. Information on the instruments is available in Thouret et al. (1998) and Marenco et al. (1998) for ozone and in Nédélec et al. (2003, 2015) for CO. The model assessment against the aircraft-based measurements is performed using the Interpol-IAGOS software (Cohen et al., 2021) that consists in projecting the IAGOS data onto the model grid and to derive monthly means for each
sampled grid cell. The model monthly averages are then derived from the daily output by applying a mask with respect to the IAGOS sampling. Last, for each data set, these monthly outputs are used to derive seasonal and annual climatologies. For this comparison we apply the methodology for comparison with the global model described in Cohen et al. (2021). Since all the four cruise-altitude levels are regularly crossed by the tropopause, each of them will contain both high-ozone (low-CO) and low-ozone (high-CO) grid cells. In order to account for
these discrepancies while deriving a mean bias, thus to avoid absolute biases in high values to govern the results, the normalized mean value shown in these graphics is derived from the Modified Normalized Mean Bias (MNMB).

In yearly means, the normalized mean value for ozone spreads from about 70% up to 125% and increases with the altitude, consistently with **Fig. 1**. It is also the case with the dependence on the season, represented by a 10%
difference at 300 hPa and a 30% difference at 180 hPa. The same scheme is reproduced between the three highest levels, i.e. a lower summertime value and a greater wintertime value, likely depending on the distance from the tropopause. It suggests that the vertical ozone gradient in the lower stratosphere is overestimated by the model. However, the ozone geographical distribution is remarkably well correlated between the simulation and the observations, with all the yearly $r$ coefficients greater than 0.92. It is especially the case at the highest levels. One
possible reason is a good representation of tropopause motions at northern mid-latitudes, which ensures a realistic proportion between stratospheric and tropospheric air masses in most grid cells. Carbon monoxide is characterized by relatively small biases at the IAGOS-coverage scale, showing a balance between regional positive and negative biases. As for ozone, the correlation increases with altitude. However, the poor correlation reported at the lowest level highlights the difficulties to simulate the upward transport of pollutants in the troposphere.
### 3.3 Model set-up

In this study, meteorological data from the European Centre for Medium-Range Weather Forecasts (ECMWF) ERA-Interim reanalysis have been used. The relaxation of the GCM winds towards ECMWF meteorology is
performed by applying at each time step a correction term to the GCM u and v wind components with a relaxation time of 2.5 hours (Hourdin and Issartel, 2000; Hauglustaine et al., 2004). The ECMWF fields are provided every 6 hours and interpolated onto the LMDZ grid. We focus this work on the impact of aircraft emissions on atmospheric composition, its future evolution, and its direct radiative forcing of climate. In order to isolate the impact of aircraft emissions, all snapshot simulations are performed under present-day climate conditions and run
for a period of ten years after a two-year spin-up. Therefore, ECMWF meteorological data for 2000-2009 are used. The perturbations are averaged over the last 3 years of the simulations.

For the baseline simulations, the anthropogenic emissions compiled by Lamarque et al. (2010), are added to the natural fluxes used in the INCA model. The ORCHIDEE vegetation model has been used to calculate off-line the biogenic surface fluxes of isoprene, terpenes, acetone and methanol as well as NO soil emissions as described by Lathière et al. (2005). $NH_3$ emissions from natural soils and ocean are taken from the the Bouwman et al. (1997). Natural emissions of dust and sea salt are computed using the 10m wind components from the the ECMWF reanalysis. For the future simulations (2050), the Representative Concentration Pathways (RCP) 6.0 anthropogenic and biomass burning emissions provided by Lamarque et al. (2011) are used. Natural emissions for both gaseous species and particles are kept to their present-day level. Lightning is an important source of $NO_x$ in the upper-troposphere at aircraft cruise altitudes. The lightning $NO_x$ emissions are parameterized in the model based on convective cloud heights as described in Jourdain et al. (2001). Based on this parameterization, the total lightning $NO_x$ emissions for the baseline simulation is 5.5 TgN/yr. The methane surface mixing ratio used for both chemistry simulations and radiative forcing calculations is fixed to 1769 and 1895 ppbv for the "present-day" (2004) and 2050 baseline simulations, respectively. For $N_2O$, the surface mixing ratio is fixed to 323 and 355 ppbv for 2004 and 2050, respectively.

The impact of aviation emissions on atmospheric composition is calculated based on a 100% perturbation methodology, comparing the results of the simulations with aircraft emissions to a reference simulation with zero aircraft emissions. As discussed by Søvde et al. (2014), in previous work, both 5% perturbations (e.g. Hoor et al., 2009; Hodnebrog et al., 2012) and 100% perturbations (e.g., Gauss et al., 2006, Søvde et al., 2014) have been applied. The latter gives the overall effect of aircraft, without considering compensating effects from other emission sectors due to chemical non-linearity (Grewe et al., 2019). This non-linear chemistry depends mostly on the background of $NO_x$ and hydrocarbons, and increases with larger perturbations in $NO_x$. In this study we focus on aircraft emissions solely in contrast to other studies which compared impact of emissions from different transport modes (e.g., Hoor et al., 2009; Koffi et al., 2010; Hodnebrog et al., 2012). Sensitivity analysis (as we envisaged our simulations) aims at characterizing the concentration change resulting from a given emissions change (Clappier et al., 2017). On the other hand, source apportionment approaches aim to quantify contributions by attributing a fraction of the pollutant concentration to each emission source (e.g., tagged species approach) (Grewe et al., 2010; Clappier et al., 2017; Grewe et al., 2019; Matthes et al., 2021). The two methods provide different results but also different information. The source apportionment accounts for non-linearities and is used to retrieve information on the source contribution to the concentration of one pollutant (e.g., contributions of different transport modes to ozone). Sensitivity or impact methods are used to determine the impact of abatement strategies. In this study which focuses on aircraft impact solely, we adopt the 100% perturbation, keeping in mind this may mask some mild non-linearities.

### 3.4 Radiative forcing calculations

Several radiative forcings associated with atmospheric composition changes due to aircraft emissions are computed on-line during the LMDZ-INCA simulations. This is in particular the case for the various components of the direct aerosol forcing. The radiative calculations in the GCM are based on an improved version of the ECMWF scheme developed by Fouquart and Bonnel [1980] in the solar part of the spectrum and by Morcrette [1991] in the thermal infrared. The shortwave spectrum is divided into two intervals: 0.25–0.68 µm and 0.68–4.00 µm. The model accounts for the diurnal cycle of solar radiation and allows fractional cloudiness to form in a grid box. These radiative forcings are calculated as instantaneous, clear-sky and all-sky forcings at the surface and at the top of the atmosphere. In section 6 the all-sky forcings at the top of the atmosphere will be presented for aerosols as was done for instance by Hauglustaine et al. (2014).

For ozone and stratospheric water vapour, a different protocol is used. For these two species, the radiative forcings at the tropopause are calculated with an off-line version of the LMDZ GCM radiative transfer model described above. In this off-line version, the forcings are calculated on a monthly mean basis using the temperature, water vapour, cloud distributions and optical properties, surface albedo, and ozone fields stored from the GCM simulations and read from pre-established history files. The fixed-dynamical heating approximation is then applied to the calculations with a thermal adjustment of the stratosphere. The radiative code iterates until the forcings at the top of the atmosphere converges with the forcings at the tropopause. The iterations are performed with a one-day timestep over 200 days. This radiative transfer model off-line of the LMDZ-INCA model has for instance already been used to calculate the tropospheric ozone radiative forcings in Berntsen et al. (2005) or more recently in Li et al. (2016).

### 4 "Present-day" baseline perturbations

#### 4.1 Aviation impact on atmospheric composition

**Figure 3** presents the daily changes in concentration associated with the base REACT4C_2006 aircraft emission inventory for several key species at 250 hPa (i.e. cruise altitude). For $NO_x$, a strong seasonal cycle is calculated with a fall-winter maximum reaching 39 pptv and a summer minimum of 15-20 pptv located at 30°-60°N and corresponding mostly to the transatlantic flight corridors. A northward transport of $NO_x$, associated with the transport of mid-latitude air masses to the polar regions is visible. During spring, the $NO_x$ mixing ratio increases





by up to 20 pptv at high latitudes. The geographical distribution depicted in **Fig. 4** shows that $NO_x$ increases by up to 60 pptv in regions with high aircraft emissions over Europe, Northern America. It also extends to North-East Asia and Japan, reaching more than 40 pptv. As a consequence of this increase in $NO_x$ concentrations, ozone increases by 3-6 ppbv at flight altitude at northern mid-latitudes. A marked seasonal cycle (**Fig. 3**), associated with the $NO_x$ increase and higher photochemical activity, is calculated at mid-high northern latitudes and peaks in May.
A maximum zonal mean ozone increase reaching almost 7 ppbv is calculated in polar regions where photochemistry is active and where mid-latitude ozone is transported and accumulates. Due to its longer lifetime, the change in ozone at 250 hPa reaches 2-4 ppbv over most of the northern hemisphere above 30°N (**Fig. 4**).

The zonal mean distributions of the $NO_x$ and $O_3$ perturbations are shown in **Fig. 5** for January and May. The $NO_x$ increase reaches a maximum of about 45 pptv at 250 hPa and at 40°N-60°N during both seasons. In May, the
mixing of air masses towards higher latitudes is visible with increased mixing ratios reaching about 25 pptv at the pole. The induced ozone perturbation ranges from a maximum of 3 ppbv in January to a maximum of 6-7 ppbv in May. These results agree with the model intercomparison of Søvde et al. (2014) who calculated, with the same aircraft emission inventory, a maximum ozone increase ranging from 4.8 to 8.8 ppbv during summer, and a lower impact in winter associated with less photochemical activity, and ranging from 3.4 to 4.4 ppbv. They are also in
agreement in terms of distribution and ozone increase with the earlier model intercomparison results by Hoor et al. (2009) obtained with the QUANTIFY_2000 emission inventory, and with the peak absolute ozone increase of 5 to 8 ppbv calculated by Olsen et al. (2013). The calculated maximum ozone perturbation occurs between 300 and 200 hPa in both seasons. A downward transport of the ozone produced is visible down to 800 hPa at 30°N-40°N. Due to higher photochemical activity at high latitudes (>60°N), and mixing and accumulation of air masses
around the pole, the ozone increase is centered in polar regions in May. In winter, this maximum is located at lower latitudes between 40°N and 60°N, as illustrated in **Fig. 4**. The maximum perturbation associated with aircraft emissions appears just above the tropopause at high latitudes showing the need to account for both chemistry in the troposphere and in the stratosphere (Gauss et al., 2006; Søvde et al., 2014; Khodayari et al., 2014a). As a result of the $NO_x$ and hence ozone increases, an increase of OH located between 30°N and 50°N, depending on the solar
radiation seasonal cycle, and reaching 14-20 $10^{-3}$ pptv is calculated (**Fig. 3**).

The increase in water vapour associated with $H_2O$ aircraft emissions shows a strong seasonal cycle and reaches a maximum of 3.5 ppbv in spring at 250 hPa (**Fig. 3**). The zonal mean distribution (**Fig. 5**) shows that the maximum is located in the stratosphere where the water vapour lifetime is longer. The increase in stratospheric water vapour
reaches 19 ppbv at 200 hPa in January at 60°N. In May the increase reaches about 10 ppbv at this latitude. This is significantly lower than the 64 ppbv annual mean maximum increase calculated by Wilcox et al. (2012) with a Lagrangian model and considering water vapour as a passive tracer. With a Lagrangian model Morris et al. (2003) also calculated an increase of water vapour due to aircraft emissions of more than 150 ppbv just above the tropopause and of less than 2 ppbv in the stratosphere. In our model set-up, the increase of water vapour is reset
to zero below the model tropopause at each time-step, strictly limiting the aircraft perturbation to a stratospheric perturbation.

For BC (**Fig. 3**), the seasonal cycle is well marked with a maximum reaching 0.16 ng/m³ in winter-spring and a minimum in summer of 0.01 ng/m³. This aerosol accumulates at cruise altitude during winter leading to a marked
maximum. In summer, due to more intense atmospheric mixing, these high concentrations are mixed to lower altitudes. Meridional and poleward transport is also visible for these particles and higher concentrations are reached from 40°N to the pole, in agreement with the transport of $NO_x$ discussed earlier. The change in $SO_2$ concentration (not shown) exhibits a similar feature with a winter maximum reaching more than 6 pptv. For this aerosol precursor, despite direct aircraft emissions, a decrease is calculated in summer reaching 3.5 pptv, and
corresponding to the $SO_2$ oxidation by OH forming sulfate particles. As a consequence of this enhanced production, a maximum increase in $SO_4$ is calculated from May to September and reaches 8-12 ng/m³ (**Fig. 3**).

**Figure 6** shows the geographical distribution of the BC perturbation at 250 hPa. In January, the maximum reaching 0.17 ng/m³ is located over source regions in Northern America, Europe and Eastern Asia with zonal transport over
the Pacific and Atlantic flight corridors. The distribution in May clearly shows transport and accumulation of BC in polar regions. The zonal mean distribution of the BC perturbation is shown in **Fig. 7** and exhibits a maximum of 0.15 ng/m³ between 300 and 200 hPa at 40°N-70°N. A redistribution of these flight altitude emissions through subsidence is visible with a secondary maximum of about 0.08 ng/m³ calculated in the lower troposphere around 30°N. These results are in line with the perturbations calculated by Righi et al. (2016) and ranging from 0.05-0.1
ng/m³ in annual mean at cruise altitude in the northern hemisphere. In the lower troposphere, Righi et al. (2016) calculated a somewhat higher BC concentration increase reaching up to 0.5 ng/m³ near the surface.

The geographical distribution of sulfates at 250 hPa (**Fig. 6**), clearly shows a strong seasonal cycle associated with increased oxidation of $SO_2$ in spring, reaching 14 μg/m³ in May over a large part of Europe and Asia. In **Fig. 7**,
the zonal mean $SO_2$ perturbation varies between 8 ppt in January and 5 ppt in May at 200 hPa between 40°N and 90°N. The associated sulfate perturbation reaches a maximum of 10 ng/m³ in May with a minimum of 5 ng/m³ simulated in January and localized, as for BC, between 300 and 200 hPa at the latitude band between 30°N-50°N and with a clear poleward transport in May. As seen for BC, a significant subsidence of the sulfate perturbation is calculated between 30 and 40°N. Again, these results agree with the perturbations calculated by Righi et al. (2016)





with the EMAC model and reaching 2-5 ng/m³ in annual mean at cruise altitude in the northern hemisphere. In the lower troposphere, Righi et al. (2016) calculated higher SO₄ concentration increase reaching up to 10 ng/m³ near the surface. Since similar emission indexes are used in both studies, this points towards a more efficient removal of aerosols in LMDZ-INCA than in EMAC.

Nitrates are not emitted by aviation but their distribution is affected by two competing processes. On one hand, the increase of SO₄ reduces the amount of NH₃ available for ammonium nitrate formation and on the other hand the increase of NOₓ enhances the production of nitrates. At cruise altitudes, the strong increase in SO₄ dominates and an overall decrease in nitrates of up to -7 µg/m³ is calculated in May over Europe and Asia, collocated with the increase in sulfates (**Fig. 6**). In regions characterized with high NH₃ concentrations, in India or south-east Asia,
enough ammonia is still present after ammonium sulfate production to increase the production ammonium nitrate when more NOx associated with aircraft emissions are present. This is in particular the case in January around 30°N in the lower troposphere (**Fig. 7**), but also at cruise altitudes in localized areas in India and South-East Asia (**Fig. 6**). In these regions an increase of nitrates reaching more than 9 ng/m³ is calculated. Righi et al. (2016) calculated a very similar zonal mean pattern for the nitrate concentration perturbation, decreasing in the upper
troposphere by up to 1 µg/m³ in annual mean and increasing by 5-10 ng/m³ in the lower troposphere. Similarly, the zonal mean perturbation pattern for nitrate aerosols agrees with the results presented by Unger (2011).

**4.2 Impact of flight altitude changes**

The atmospheric lifetime of pollutants emitted by aviation is highly dependent on the altitude at which they are injected into the atmosphere (Grewe et al., 2002; Gauss et al., 2006; Fröming et al., 2012; Søvde et al., 2014; Matthes et al., 2017). The sensitivity of the calculated perturbations to the flight altitude is illustrated in this section based on the REACT4C_PLUS and REACT4C_MINUS emissions corresponding, respectively, to an increase or decrease of the flight cruise altitude by 2000 feet compared to the REACT4C_2006 baseline inventory.
**Figure 8** shows the impact of the flight altitude variation on the zonal mean distribution of key species compared to the baseline scenario. These variations are illustrated for May conditions when the maximum impact of aircraft emissions is calculated by the model, as illustrated in the previous section. As expected as a consequence of the chemical lifetime increase with altitude, a higher (resp. lower) flight cruise altitude increases (resp. decreases) the
change in ozone mixing ratio by 1.7 ppbv (resp. -1.6 ppbv) between 150 and 250 hPa compared to the baseline scenario. The impact on ozone is comparable to the results obtained by Søvde et al. (2014) and Matthes et al. (2017; 2021) (1-2 ppb in summer and 0.4-1 ppb in winter). Similarly, the BC concentration increases by 0.031 ng/m³ when the flight altitude is increased and, in contrast, decreases by 0.032 ng/m³ with a lowered flight altitude.

Directly related to the response of SO₂ to flight altitude changes, the concentration of sulfates shows a behavior similar to the primary aerosol BC, and increases by a maximum of 2.3 ng/m³ between 250 and 150 hPa at 40°-90°N in the REACT4C_PLUS case and decreases by 2.3 ng/m³ in the REACT4C_MINUS simulation. In contrast, the variation of nitrate aerosols shows an opposite behavior associated with the change in sulfates. An increase in sulfates reduces the NH₃ available for forming ammonium nitrates particles in favour of ammonium sulfates and
NO₃ decreases by 0.84 ng/m³ at flight altitude in the REACT4C_PLUS simulation. On the other hand, the decrease in sulfate concentration calculated in the REACT4C_MINUS scenario induces an increase of ammonium nitrate particles of 0.83 ng/m³ at 200 hPa. These sensitivity simulations show that changing the aircraft flight altitude has a marked impact on the ozone and aerosol responses to emissions (of about 25% compared to the baseline simulation in May) and hence on the associated radiative forcings, as will be analyzed in the section 6.

**5 Future impact of aviation**

**5.1 Future baseline scenario**

In addition to these simulations using the REACT4C_2006 emission inventory, a set of future simulations using the QUANTIFY emission inventories have been carried out for the year 2050. The corresponding distributions of the baseline perturbations for the QUANTIFY_2000 inventory are in line with the results obtained for REACT4C_2006 (**Fig. S1**). However, in the case of the QUANTIFY_2000 emissions, due in particular to the higher fuel consumption, the maximum perturbations are generally slightly higher. In zonal mean, these
perturbations reach 6.8 ppbv for ozone in May between 250 hPa and 350 hPa, 0.16 ng/m³ for BC, 11.3 ng/m³ for SO₄, and -3.95 ng/m³ for NO₃, to be compared with 6.4 ppbv, 0.15 ng/m³, 10.2 ng/m³ and -3.71 ng/m³, respectively, in the case of REACT4C_2006 (see Section 4).

For future simulations (2050), the baseline simulation corresponds to the aviation emissions from the
QUANTIFY_A1 inventory. **Figure 9** illustrates the perturbation associated with aircraft emissions for this scenario for key constituents (May as the seasonal maximum of the perturbation). The NOₓ mixing ratio increases in the upper troposphere by up to 194 pptv in January at 200 hPa and 170 pptv in May (not shown). As a consequence, ozone increases from 9.2 ppbv in January to up to 19.6 ppbv in May at flight altitude. These values are comparable to the values calculated by Søvde et al. (2007) (7 to 18 ppbv for monthly averages in January and
May respectively) but higher than the ozone increase calculated by Koffi et al (2010) (10 ppbv in July) and the





model mean given by Hodnebrog et al. (2012) for the same emission inventory. We note however that only a few models used in Hodnebrog et al. (2012) included an interactive chemisty in the stratosphere, hence imposing ozone to climatologies or calculated with simplified parameterizations in this region. This could have the consequence to dampen the response of ozone in the upper-troposphere. This finding is confirmed by the higher tropospheric ozone increase associated with aircraft $NO_x$ emissions calculated by global models including both interactive chemistry in the troposphere and in the stratosphere (Olsen et al., 2013; Khodayari et al., 2014a; Brasseur et al., 2016; Skowron et al., 2021). The increase in BC is also significant and reaches 0.6 ng/m³ in January and 0.55 ng/m³ in May, a factor of 3 larger than the QUANTIFY_2000 perturbation. As noted earlier, in addition to the maximum increase at flight altitude, a secondary maximum reaching 0.3 ng/m³ and associated with the downward redistribution of aircraft emissions is also calculated at ground level. The increase in $SO_4$ for this scenario reaches 15.6 ng/m³ in January and 32.8 ng/m³ in May. The marked increase in $SO_4$ concentrations at cruise altitude is responsible for a subsequent decrease in $NO_3$ reaching -9.4 ng/m³ and -21 ng/m³ in May. Below about 500 hPa, an increase in $NO_3$ concentrations reaching 14 ng/m³ in May and as much as 47 ng/m³ in January is simulated. These increased surface concentrations in 2050 are mostly associated with much higher $NH_3$ background concentrations at the surface in the future (in particular in South East Asia) (Hauglustaine et al., 2014), and subsequent enhanced $NO_3$ formation due to redistribution of aircraft $NO_x$ emissions to lower levels forming ammonium nitrates. This result agrees with the findings of Righi et al. (2016) and Unger (2011) who calculated a decrease of nitrate particles in the $NH_3$-limited mid-upper troposphere due to aircraft emissions and an increase in the lower troposphere and at surface. This is however in contrast to the results of Unger et al. (2013) who calculated an increase of $NO_3$ associated with aircraft emissions in most of the troposphere with a moderate 2-3% decrease in the upper-troposphere and lower-stratosphere, and consequently derived a strong negative forcing by nitrate particles as reported also by Brasseur et al. (2016).

## 5.2 Mitigation scenarios

Two alternative future scenarios are derived from the future scenario QUANTIFY_A1. As described in Section 2, the QUANTIFY_LowNOx scenario simulates the impact of $NO_x$ emissions reduced by a factor 2, mimicking a significant improvement of the engine combustion technology with respect to $NO_x$ emission index. The QUANTIFY_A1_Desulfurized scenario simulates the impact of a desulfurized fuel. In addition, two alternative scenarios representing two aircraft emission mitigation trajectories are used, these are the QUANTIFY_B1 and QUANTIFY_B1_ACARE scenarios described by Owen et al. (2010) and summarized in Section 2.

**Figure 10** shows the impact of the LowNOx and Desulfurized future emission scenarios on the zonal mean distribution of key atmospheric species. The low NOx emissions have a significant impact on the ozone increase. In this case, the zonal mean ozone increase reaches a maximum of 5.9 ppbv in January (not shown) and 12 ppbv in May at 200 hPa. This corresponds to a reduction of 3 ppbv in January and 7 ppbv in May compared to the QUANTIFY_A1 scenario (**Fig. 9**). In this scenario, $SO_4$ is only moderately affected at flight altitude and the maximum $SO_4$ increase in May remains close to 33 ng/m³. However, in the lower troposphere, less sulfates are produced in the aqueous phase at lower $O_3$ concentrations especially in subsidence regions. The increase in $SO_4$ at the surface is significantly reduced and decrease from about 17 ng/m³ at 30°N in the QUANTIFY_A1 scenario to less than 10 ng/m³ in the QUANTIFY_LowNOx simulation. In addition, as shown in **Fig. 10**, the impact on $NO_3$ is very limited at the flight cruise altitude. However, in the lower troposphere, the formation of nitrates through the reaction with $NH_3$ is decreased at lower $NO_x$ emissions and the $NO_3$ concentration increase at the surface reaches 7 ng/m³ to be compared with the 14 ng/m³ calculated in the QUANTIFY_A1 simulation (**Fig. 9**). This is a factor of 2 reduction, linear with the decrease in total $NO_x$ aircraft emissions in this scenario. This LowNOx scenario has a significant impact on air quality reducing both tropospheric ozone but also the concentration of sulfates and nitrates in the lower troposphere.

In comparison to the QUANTIFY_A1 baseline scenario, the QUANTIFY_A1_Desulfurized simulation has, as expected, a significant impact on the $SO_2$ aircraft perturbation and consequently on the formation of $SO_4$. As seen from **Fig. 10**, since no emission of $SO_2$ and $SO_4$ are considered in this simulation, the $SO_4$ increase reaching 13 ng/m³ in May around 30°N and at 300-200 hPa is only associated with the increased production from $SO_2$ oxidation at higher ozone and OH concentrations. As a consequence of the reduced change in $SO_4$, a very moderate decrease in nitrates reaching only -6 ng/m³ is calculated at flight altitude. In the lower troposphere, the increase in $NO_3$ associated with production of ammonium nitrate from surface $NH_3$ emissions and aircraft $NO_x$ reaches 14 ng/m³ as also calculated in the QUANTIFY_A1_Desulfurized scenario. These results agree with Unger (2011) and Kapadia et al. (2016) who calculated a similar impact of desulfurized fuel, a much lower increase of sulfates and nitrates at flight altitude and a significant increase in nitrates in the lower troposphere. As in Unger (2011), but in contrast to earlier work by Pitari et al. (2002), little effect of sulfate aerosols on ozone via heterogeneous chemistry is predicted in the upper troposphere. The impact of aircraft desulfurized fuel is close to the regular fuel ozone perturbation.

The results for the alternative economic and technological scenarios QUANTIFY_B1 and QUANTIFY_B1_ACARE (Owen et al., 2010) are illustrated in **Fig. 11**. QUANTIFY_B1 is a mitigation scenario for which improving air quality is a primary objective and therefore the reduction of $NO_x$, $SO_2$ and BC emissions are relatively important compared to the base case future scenario QUANTIFY_A1. In this case, as a consequence



of lower NO$_x$ emissions, the O$_3$ perturbation is reduced by more than a factor of 2 compared to QUANTIFY_A1 and the ozone zonal mean increase reaches a maximum of 4.5 ppbv in January (not shown) and of 8.6 ppbv in May compared to 9.2 ppbv and 19.6 ppbv in January and May, respectively, for the QUANTIFY_A1 simulation.

For this mitigation scenario Hodnebrog et al. (2011) derived a model mean ozone increase at cruise altitude of 3 ppbv in January and about 5 ppbv in July, somewhat lower than our results. Again, it should be kept in mind that none of the models used in Hodnebrog et al. (2012) included an interactive chemisty in the stratosphere. The BC perturbation reaches a maximum of 0.35 ng/m$^3$ at flight altitude, to be compared with 0.55 ng/m$^3$ in the QUANTIFY_A1 simulation. The sulfate concentrations increase in May by up to 22 ng/m$^3$ and as a consequence

of this SO$_4$ increase, NO$_3$ are reduced by -14 ng/m$^3$ at flight altitude, compared to 33 ng/m$^3$ and -22 ng/m$^3$, respectively, for the QUANTIFY_A1 scenario. The alternative scenario QUANTIFY_B1_ACARE is the most ambitious mitigation future scenario used in this study with very strong emission reductions. This scenario is an "extreme" scenario to limit the environmental impact of aviation. In this scenario, the NO$_x$ emissions are reduced by almost a factor of 5 and the ozone increase reaches in May a maximum of 6.2 ppbv, a value even lower that the

ozone increase calculated under the QUANTIFY_2000 (6.8 ppbv) and REACT4C_2006 (6.4 ppbv) simulations. A significantly attenuated aerosol increase is also calculated for the QUANTIFY_B1_ACARE scenario, and BC and SO$_4$ increase by up to 0.25 ng/m$^3$ and 16 ng/m$^3$ at flight altitude respectively. The nitrate concentration decreases by 10 ng/m$^3$ at flight altitude and increases by up to 3.8 ng/m$^3$ in the lower troposphere (**Fig. 11**).

**6 Radiative forcing of climate**

**6.1 Changes in ozone burden and methane lifetime**

**Table 3** summarizes the impact of aircraft emissions on the global burden of ozone and on the methane lifetime.
Ozone increases by 4.7 Tg (0.430 DU) for the REACT4C_2006 reference scenario and an ozone production efficiency of 6.6 TgO$_3$/TgN is derived. The ozone sensitivity to aircraft NO$_x$ emissions (0.6 DU/TgN) is similar to the adopted mean value reported by Holmes et al. (2011) (**Table S4**) and in the higher range of the 0.39-0.63 DU/TgN model range provided by Myhre et al. (2011). A somewhat higher values (0.79 DU/TgN for present-day conditions) was derived by Khodayari et al. (2014b) with the AEDT inventory. We note however that the ozone
burden calculated by Koffi et al. (2010), presented in a model intercomparison by Hoor et al. (2009) (model mean of 5.6 TgO$_3$/TgN) and by Myrhe et al. (2011) were for global models including tropospheric chemistry solely. In most of these earlier models (and in particular in an earlier 19 level version of LMDZ-INCA) the stratospheric chemistry was not included and ozone constrained to climatogies above a potential temperature of 380 K. This neglects the ozone change above the tropopause, as seen from **Fig. 5**. Olsen et al. (2013) derived a high model
spread for the ozone burden change per emitted NO$_x$ of 2.5-11 TgO$_3$/TgN with some linearity of the model responses to the NO$_x$ emission. At higher flight altitudes (REACT4C_PLUS), the increase in ozone is about 15% higher than for the reference case and at lower cruise altitude (REACT4C_MINUS) the increase in ozone is 11% lower, reflecting longer NO$_x$ residence time at higher altitude. This results in an increase in ozone production efficiency reaching 7.5 at higher flight altitudes and a decrease to 5.8 at lower flight cruise altitude. In the case of
QUANTIFY_2000, the ozone burden increases by 5.0 Tg (0.457 DU) and the ozone production efficiency is slightly lower (6.0) than for the REACT4C_2006 inventory, reflecting aircraft emissions deposited at somewhat lower flight altitudes (Skowron et al., 2013). The reference present-day methane chemical lifetime calculated with this model version is 10.6 yr, close to the present-day (2000) value reported by Voulgarakis et al. (2012) for a previous version of the LMDZ-INCA model. Due to the increase in OH concentration when NO$_x$ aircraft emissions
are considered, based on the REACT4C_2006 emission scenario, the methane chemical lifetime decreases by -1.0% (-1.15% for the QUANTIFY_2000 aircraft emissions). A somewhat higher values (-1.65%/TgN for present-day conditions) was derived by Khodayari et al. (2014b) with the AEDT inventory.

For the 2050 scenarios, ozone increases by up to 20.3 Tg (1.86 DU) for the QUANTIFY_A1 scenario and by 7-8
Tg (0.64-0.73 DU) for the B1 and B1_ACARE scenarios. Despite the very different atmospheric background in 2050, the derived ozone production efficiency for the A1 scenario (6.14) is rather close to the QUANTIFY_2000 reference value (6.0). A higher ozone production efficiency of respectively 7.7 and 7.9 is derived in the case of the B1 and B1_ACARE scenarios. A similar feature was derived by Hodnebrog et al. (2011; 2012) for the QUANTIFY inventories in 2050, with ozone production efficiencies averaged among several models (and for
different background conditions) of respectively 5.7, 7.4, 7.9 for the A1, B1 and B1_ACARE scenarios. For the A1_LowNOx scenario (future NO$_x$ emissions reduced by 50%), the ozone burden decreases by 56% compared to the A1 scenario, showing a very slight non-linearity. We calculate a methane reference lifetime in 2050 of 11.4 yr. Due to aircraft emissions, the future methane lifetime decreases by 4.7% for the QUANTIFY_A1 scenario and by 1.8% and 1.2% for the B1 and B1_ACARE scenarios respectively. The calculated future ozone changes can
also be compared with more recent estimates involving global models including both tropospheric and stratospheric chemistry. With the ACCRI_2050_Base and 2050-S1 emissions (**Table S2**), Olsen et al. (2013) derived a global ozone burden increase ranging from 10.3 to 18.6 Tg (2.6-7.24 TgO$_3$/TgN) and from 4.5 to 13.8 Tg (2.9-8.8 TgO$_3$/TgN), respectively. More recently, Skowron et al. (2021) calculated with the MOZART-3 model and based on a high (5.59 TgN) and a low (2.17 TgN) future (2050) aircraft emission scenarios, global ozone
increases of respectively 31 Tg and 15.4 Tg with some variation depending on the surface emission scenario, providing respectively 5.5 TgO$_3$/TgN and 7.1 TgO$_3$/TgN. Our results agree with the ozone sensitivity to NO$_x$ and all these future simulations point to a decreasing ozone production efficiency at higher NO$_x$ emissions.

### 6.2 Radiative forcing from NOx emissions


The calculated radiative forcing of climate for the different scenarios are summarized in **Table 4**. The increase in tropospheric ozone due to the aircraft NOx emissions is responsible for a positive radiative forcing F$_{O3}$ of 15.9 mW/m$^2$ in the case of the reference scenario REACT4C_2006. This forcing increases by 1.85 mW/m$^2$ (12%) when the flight altitude is increased in the case REACT4C_PLUS and decreases by 1.59 mW/m$^2$ (10%) when the flight

altitude is decreased in the case REACT4C_MINUS. In the case of QUANTIFY_2000 the calculated ozone forcing is 17.2 mW/m$^2$, a value somewhat higher than REACT4C_2006 due to higher NOx emissions. This value agrees with the ozone forcings calculated by Hoor et al. (2009) in a multi-model study who derived a mean forcing of 16.3 mW/m$^2$ for the QUANTIFY_2000 scenario. The calculated ozone forcing provides a forcing relative to the NOx emissions of 22.35 mW/m$^2$/TgN. This value is in good agreement with Myhre et al. (2011) who calculated

a range for five models of 16-25 mW/m$^2$/TgN and Holmes et al. (2011) who calculated based on factor decomposition a forcing of 21.6 ± 7.2 mW/m$^2$/TgN. As noted by Skowron et al. (2013), the ozone forcing is very much dependent on the NOx emission vertical profile with normalized forcings ranging from 16.5 to 20.5 mW/m$^2$/TgN in their MOZART-3 simulations depending on the aircraft emission inventory considered. More recently, Lee et al. (2021) derived a best estimate value of 25.1±7.2 mW/m$^2$/TgN. As obtained by Holmes et al.

(2011), we derive an ozone forcing per DU of 37.1 mW/m$^2$/DU, in agreement with this previous work (**Table S4**). With different emission inventories, Olsen et al. (2013) reported an even wider range for the normalized ozone forcing of 6-37 mW/m$^2$/TgN. For the future scenarios, the tropospheric ozone forcing reaches 70.6 mW/m$^2$ for the QUANTIFY_A1 scenario and 27.6 and 18.7 mW/m$^2$ in the case of the B1 and B1_ACARE mitigation scenarios, respectively. These values agree with the ozone forcings calculated by Hodnebrog et al. (2011; 2012) in a multi-

model study who derived forcings of 61.3±14.0 mW/m$^2$, 24.1±7.7 mW/m$^2$ and 18.9±6.6 mW/m$^2$ for the QUANTIFY_A1, QUANTIFY_B1 and B1_ACARE, respectively. For the reduced NOx emission scenario (A1_LowNOx), the ozone forcing is 55% of the A1 scenario value, in line with the change in ozone burden.

The total methane forcing F$_{CH4}$ can be broken down into four distinct forcings (**Table 5**). As described in previous

work (e.g., Fuglestvedt et al., 1999; Skowron et al., 2013), the methane decrease associated with enhanced oxidation by OH, is responsible for a long-term methane direct radiative forcing that we calculate at equilibrium based on the change in the methane photochemical lifetime, and including the methane feedback on its own lifetime. According to this methodology the steady-state methane mixing ratio decrease due to aircraft emissions is given by (Prather, 1994; Fiore et al., 2009; Holmes, 2018):


$$q_{CH4} = q^0_{CH4}\,(1 + f\,\Delta\tau_{CH4})\qquad\qquad\qquad(1)$$

where $q_{CH4}$ is the new steady-state methane mixing ratio, $q^0_{CH4}$ is the reference methane mixing ratio (taken as 1769 ppbv for the "present-day" and 1985 ppbv in 2050), $f$ denotes the methane feedback on its own lifetime

(Prather et al., 2001; Holmes et al., 2011), and $\Delta\tau_{CH4}$ (%) is the change in the methane lifetime due to aircraft emissions and subsequent OH perturbation (**Table 3**). The associated direct radiative forcing of climate of this change in methane is calculated using the simple formula revised by Etminan et al. (2016). The inclusion of the shortwave contribution in this revised expression results, for instance, in an increase by 24.5% of the methane forcing associated with a halving of its concentration, compared to the previous literature (Myhre et al., 1998). For

the methane feedback factor, $f$, Prather et al. (2001) derived a best estimate value of 1.4 based on a multimodel range of 1.33–1.45. A similar best estimate value was recommended by Holmes et al. (2011). Fiore et al. (2009) derived a multimodel ensemble mean of 1.33 and a range of 1.25-1.43, including a value of 1.31 contributed with an earlier version of the LMDZ-INCA model. The feedback factor has been recalculated in this study for the current version of our model based on a reference simulation and a 10% perturbation in methane surface mixing

ratio simulation. Based on this set of simulations and the methodology described in Fiore et al. (2009), we recalculated a methane feedback factor $f$ of 1.36, in agreement with the previous estimates. This factor is then used in (1) in order to derive the change in methane mixing ratio due to aircraft emissions and the associated radiative forcing of climate. It should be noted that this estimate of the methane feedback is based on fixed surface CH$_4$ concentrations. Other methodologies have also been proposed to evaluate this feedback and in particular the use

of methane surface fluxes (Khodayari et al., 2015; Pitari et al., 2016). The fixed surface concentration perturbation method has been widely used in previous work and was found to be accurate to within 10% by Khodayari et al. (2015). We calculate a direct radiative forcing at equilibrium associated with this methane decrease F$_{CH4-OH}$ of - 11.0 mW/m$^2$ for the "present-day" REACT4C_2006 simulation (-12.5 mW/m$^2$ for the QUANTIFY_2000 simulation). For future conditions, the direct methane forcing ranges from -13.3 mW/m$^2$ for the

QUANTIFY_B1_ACARE to -52.8 mW/m$^2$ for the QUANTIFY_A1 scenario (**Table 5**).

The indirect forcings associated with this change in methane mixing ratio through long-term tropospheric ozone and stratospheric water-vapour ajustments were recalculated with the LMDZ-INCA global model. This was done by imposing the new methane steady-state surface mixing ratio $q_{CH4}$ (1744 ppbv) calculated from (1) for the

REACT4C_2006 simulation in the 3D model and running for a period of 10 years in order to determine the associated change in ozone and stratospheric water vapour by comparing to a reference simulation using $q^0_{CH4}$ (1769 ppbv) as methane surface mixing ratio. In this case, we calculated a change in ozone after 10 years of -0.93 Tg (0.085 DU) and a change in stratospheric water vapour of -2.12 Tg. We then calculated for these perturbations,





an ozone radiative forcing of -2.82 mW/m² and a stratospheric water vapour radiative forcing of -0.65 mW/m² (**Table 5**). This provides an indirect long-term ozone forcing of 116 mW/m²/ppmv of $CH_4$ and an indirect long-term stratospheric water vapour forcing of 27 mW/m²/ppmv of $CH_4$. These numbers were then used to derive the indirect long-term ozone and water vapour forcings for the other aircraft emission scenarios based on a simple scaling with the methane mixing ratio change calculated with Eq. (1) for all simulations. For the future simulations we calculated an ozone long-term forcings $F_{CH4-O3}$ ranging from -3.6 mW/m² (B1_ACARE) to -13.96 mW/m² (A1)

and a stratospheric water vapour forcings $F_{CH4-SWV}$ ranging from -0.82 mW/m² (B1_ACARE) to -3.22 mW/m² (A1).

Carbon dioxide is the end-product of the methane atmospheric oxidation. The production of $CO_2$ is hence an indirect radiative forcing resulting from the change in methane mixing ratio due to increased oxidation by OH

resulting from aviation $NO_x$ emissions. Since intermediate carbon containing methane oxidation products are subject to dry and wet deposition, not every oxidized methane molecule results in a produced $CO_2$ molecule. In this study, we assume that 1 mole of change in $CH_4$ oxidation leads to 0.6 mole of $CO_2$ produced (Folberth et al., 2005; Boucher et al., 2009). The decrease in $CH_4$ due to enhanced oxidation is then translated into a change in $CO_2$ and converted to a radiative forcing based on the simple formula from Myhre et al. (1998). Based on the

REACT4C_2006 change in methane mixing ratio we derive an indirect $CO_2$ forcing $F_{CH4-CO2}$ of 8.25 mW/m²/ppmv of $CH_4$. This normalized number is used to calculate the forcings for the other simulations. These forcings are -0.20 mW/m² and -0.23 mW/m² for the "present-day" simulations REACT4C_2006 and QUANTIFY_2000, respectively, and range from -0.21 mW/m² (B1_ACARE) to -0.81 mW/m² (A1) in 2050 (**Table 5**).

The sum of these four components provides $F_{CH4}$ the total methane radiative forcing associated with aircraft $NO_x$ emissions. This total forcing is -14.7 mW/m² and -16.7 mW/m² for the "present-day" REACT4C_2004 and QUANTIFY_2000 inventories, respectively. In 2050, the total methane forcing ranges from -17.9 mW/m² for QUANTIFY_B1_ACARE to -70.8 mW/m² for the A1 scenario (**Table 5**). The change in $CH_4$ mixing ratio itself represents 75% of this total methane forcing. The indirect changes through long-term tropospheric ozone,

stratospheric water vapour and oxidation to $CO_2$ contribute for 19%, 4% and 1%, respectively.

As seen in **Table 4**, the $NO_x$ forcing components resulting from the tropospheric ozone positive forcing and the methane negative forcing largely offset each other resulting in a slightly positive forcing of 1.2 and 0.5 mW/m² for the REACT4C_2006 and QUANTIFY_2000 "present-day" simulations. In 2050, a net negative forcing of -

0.28 mW/m² is calculated for the QUANTIFY_A1 scenario. For the alternative scenarios B1 and B1_ACARE we still derive positive forcings of 1.14 and 0.84 mW/m². This partial compensation and the resulting positive or negative $NO_x$ net forcing depends on the level of $NO_x$ present in the background atmosphere and the location of emissions (Stevenson and Derwent, 2009; Gilmore et al., 2013; Skowron et al., 2021). Thus, for the future, higher concentrations of $NO_x$ increase the OH response and the effect of methane so that total $NO_x$ forcings even become

negative. **Table S3** compares the methane forcings calculated with the revised $CH_4$ radiative forcing parameterization proposed by Etminan et al. (2016) and by the former parameterization (Myhre et al., 1998). The net $NO_x$ negative forcings calculated for future conditions in this study are largely affected by this methane forcing parameterization, since the methane total forcing is higher by about 15% when Etminan et al. (2016) parameterization is used. In the case of the methane forcing former parameterization (Myhre et al., 1998), positive

$NO_x$ net forcings ranging from 3.3-5.0 mW/m² for "present-day" conditions to 3.3-9.7 mW/m² in 2050 are calculated. A similar conclusion regarding the possibility to have negative $NO_x$ net forcings has been reached by Skowron et al. (2021). The radiative forcings associated with $NO_x$ emissions are decomposed into its various factors following Holmes et al. (2011) methodology, and compared to previous estimates at **Table S4.** A good agreement is found for the various components or sensivities of the chemistry and of the radiative forcings with

the previous estimate provided by Holmes et al. (2011). The major difference arises from the methane forcing calculation based on Etminan et al. (2016) in our study. Similarly, the forcings normalized by emissions are also generally in agreement with Lee et al. (2021). Including the forcings associated with particles ($SO_4$ and $NO_3$) is found to change the sign of the next $NO_x$ radiative forcings compared to these earlier studies.

The sensitivity of the $NO_x$ associated forcings with aircraft flight altitude has been investigated in earlier studies (Gauss et al., 2006; Frömming et al., 2012; Søvde et al., 2014). In particular, based on the REACT4C emission inventories, Søvde et al. (2014) derived, in a multi-model study, an increase of the total $NO_x$ forcing of 2±1 mW/m² for aircraft cruising at higher altitudes and a decrease of the forcing by the same value for aircraft cruising at lower altitudes with a change essentially due to the ozone forcing. In this study, the $NO_x$ net forcing is increased by 1.74

mW/m² for aircraft cruising at higher altitudes and decreased by 1.58 mW/m² at lower altitudes. These values agree with this previous estimate. It should be noticed that based the Etminan et al. (2016) parameterization, the $NO_x$ net forcing turns from positive to negative when the aircraft cruise altitude is reduced (**Table S3**).

As illustrated by the REACT4C_2006_NOx_Only simulation results (only the $NO_x$ emissions are perturbed by aviation without including the other chemical and aerosol emissions) shown in **Table 4** and **Tables S4**, additional forcings arise from aircraft $NO_x$ emissions. This sensitivity simulation shows that the change in OH associated with $NO_x$ emissions is responsible for an increased formation of sulfate particles and a negative forcing from these aerosols of -2.0 mW/m². Similarly, $NO_x$ emissions and concentrations increase the formation of nitrate particles, also responsible for an additional negative forcing of -0.12 mW/m². The nitrate forcing calculated by Pitari et al.





(2016) is much higher (-1.7 mW/m²) but the role played by ammonia was not considered in this earlier study. The ammonium cycle is important to consider since we discussed earlier that in the lower troposphere increased $NO_x$ from aircraft emissions increase nitrates particles. However, in the upper troposphere, increased sulfate concentrations will favor the titration of ammonia to form ammonium sulfates leading to lower ammonium nitrate concentrations despite the increased $NO_x$ and $HNO_3$ concentrations. The total indirect forcing of sulfate and nitrate

aerosols associated with $NO_x$ emissions is -2.12 mW/m². When these aerosol direct radiative forcings are considered, the total net $NO_x$ forcing switches from a positive value of 1.19 mW/m² from tropospheric ozone plus methane to -0.93 mW/m². When these indirect terms are accounted for, we derive a negative net $NO_x$ radiative forcing per TgN emitted by aviation as summarized in **Table S4**.

It should also be noted that these two forcings from ozone and methane represent perturbations at steady-state. This assumption is correct for tropospheric ozone which has an averaged of a few weeks but is not for methane which has lifetime of 10.6 years in our model. For methane, the response does not reach a steady-state in any given year and the response in a particular year depends on the historical time-evolving emissions. This limitation was for instance discussed and illustrated by Khodayari et al. (2014b), Grewe et al. (2019), Lee et al. (2021) and

Matthes et al. (2021). For this reason, a transient correction factor was applied to the total methane forcing by Myhre et al. (2011), Hodnebrog et al. (2011; 2012), Matthes et al. (2021) and in the best estimate provided by Lee et al. (2021). However, this factor has not been considered in Brasseur et al. (2016) or, more recently, in Skowron et al. (2021). The major difficulty with this non-steady state factor is that its determination is strongly model dependent. Due to the long time-integration needed to investigate the methane response, complex models have not

been used so far and the non-steady state factor has been determined based on simplified or parameterized chemistry-climate models (Grewe and Stenke, 2008; Lee et al., 2021). Another difficulty in determining this factor, and in particular for future scenarios, arises from its dependency on the considered year or on the assumed future emission pathway. For the QUANTIFY_2000 emissions, Myhre et al. (2011) derived, based on Grewe and Stenke (2008) methodology, a non-steady state correction factor of 0.65. A similar factor has been used by Matthes et al.

(2021). This factor has been recently reassessed by Lee et al. (2021) with a two-dimensional model and a factor of 0.73 and 0.79 were determined for 2000 and 2018, respectively. For the future 2050 scenarios, Hodnebrog et al. (2011; 2012) derived factors of 0.74, 1.0, and 1.15 for the QUANTIFY A1B, B1 and B1_ACARE, respectively. Considering the uncertainty associated with this factor, the forcings calculated in this study assumed a steady-state methane perturbation, as was done by Skowron et al. (2021). However, for sake of completeness, **Table S5**

compares the standard steady-state forcings and the forcings with non-steady factors applied. The factors provided by Lee et al. (2021) for the 2000 and 2004 simulations and by Hodnebrog et al. (2011; 2012) for the 2050 scenarios have been used. When these non-steady methane forcings are considered, the future $NO_x$ net forcings are generally higher and even switch back to positive in 2050. We note however the high uncertainty on these factors and the need for future work in order to provide a more robust estimate of this non-steady state correction and overall

methodology.

**6.3 Direct radiative forcing from $SO_2$ and particulate carbon emissions**

The concentrations of several types of particles are perturbed by aircraft emissions and hence provide direct

radiative forcings (**Table 4**). Direct emissions of soot particles induce a positive radiative forcing of 0.46 mW/m² in the case of the REACT4C_2006 inventory and 0.54 mW/m² for the QUANTIFY_2000 inventory. With the REACT4C inventory, Pitari et al. (2015) calculated a higher direct soot forcing of 0.78 mW/m². We note however that the soot radiative forcing is subject to large uncertainties. In 2050, this forcing reaches 1.88 mW/m² for the QUANTIFY_A1 scenario. For these particles we derive a normalized forcing of 108-115 mW/m²/Tg for present-

day conditions, in the range provided by Balkanski et al. (2010) of 8-140 mW/m2/Tg and calculated based on the QUANTIFY emissions. The calculated normalized forcing is also in good agreement with the recent study of Brasseur et al. (2016) providing a range of 101-168 mW/m²/Tg.

Sulfates particle concentrations are also perturbed by the aviation $SO_2$ emissions and by the direct $SO_4$ emissions.

The negative forcing associated with an increase of these particles largely dominates the effect of the other particles and ranges from -3.87 mW/m² and -4.33 mW/m² for the REACT4C_2006 and QUANTIFY_2000 "present-day" simulations to -13.7 mW/m² in 2050 under the QUANTIFY_A1 scenario. With the REACT4C inventory, Pitari et al. (2015) calculated a direct sulfate forcing of -3.5 mW/m², a value close to this study. For these particles we derive a normalized direct forcing of -52.8 (REACT4C_2006) and -49.3 (QUANTIFY_2000) mW/m²/TgS for

"present-day" conditions, in the range provided by Unger (2011) of -56 mW/m²/TgS and Brasseur et al. (2016) of -27/-62 mW/m²/TgS. The atmospheric distribution of ammonium nitrate particles is indirectly affected by aviation emissions and dominated by the increase in sulfates favoring the formation of ammonium sulfates over ammonium nitrates. The decrease in nitrate particles is responsible for a positive forcing ranging from 0.07 mW/m² for the "present-day" to 1.3 mW/m² in 2050 for the A1 scenario. The impact of aircraft emissions on nitrates very much

depends on the $NH_3$ and $SO_2$ levels in the upper troposphere as discussed by Unger et al. (2013) who calculated a much larger nitrate forcing from aircraft in their future simulations. The results of LMDZ-INCA showed a good comparison with other models as illustrated by the model intercomparison presented by Bian et al. (2017). A slight negative forcing reaching -0.16 mW/m² in 2050 is associated with the emission of organic carbon by aviation. In total, aerosols are responsible for a negative direct forcing largely dominated by sulfate particles and ranging from

-3.4 mW/m² in the case of REACT4C_2006 to -10.6 mW/m² in 2050 for the A1 scenario.





The variation of aircraft flight altitude modifies the residence time of particles in the atmosphere. A longer life time is associated with a higher injection altitude and therefore influences concentration changes and the radiative forcing associated with these particles. The forcing associated with soot particles increases from 0.46 mW/m$^2$ in the case of the REACT4C_2006 scenario to 0.49 mW/m$^2$ at higher flight altitudes and decreases to 0.44 mW/m$^2$ at lower altitudes. The dominant forcing from aviation-induced particles remains the negative forcing from sulfates. This forcing decreases from -3.87 mW/m$^2$ to -4.38 mW/m$^2$ at higher flight altitudes and increases to -3.40 mW/m$^2$ at lower altitudes. The total aerosol forcing is dominated by sulfates and its variation with flight altitude reflects the variation of the sulfate negative forcing. The total aerosol direct forcing decreases by to -3.0 mW/m$^2$ (-12%) at lower flight altitudes (REACT4C_MINUS) and increases to -3.8 mW/m$^2$ (+13%) at higher flight altitudes (REACT4C_PLUS).

For the future QUANTIFY_A1 simulation, the use of desulfurized fuel reduces the sulfate direct forcing by 55%. The remaining sulfate forcing is associated with changes in OH and increased sulfate formation. However, the reduced direct $SO_2$ (and $SO_4$) emissions have the consequence to decrease the sulfate forcing to -6.5 mW/m$^2$. The use of desulfurized fuel also has an impact on nitrates particles. In this case, the nitrates are essentially affected by increased $NO_x$ concentrations and a negative forcing of -0.8 mW/m$^2$ is calculated. In total, the use of desulfurized fuel decreases the forcing of particles from -12.4 mW/m$^2$ to -5.6 mW/m$^2$. The ozone forcing is only slightly affected by the removal of $SO_2$ aircraft emissions in agreement with Unger (2011).

### 6.4 Radiative forcing from water vapour emissions

The radiative forcing associated with water vapour increase in the stratosphere due to direct aircraft emissions and calculated with the LMDZ-INCA model is 0.13-0.16 mW/m$^2$ for "present-day" conditions (REACT4C_2006 and QUANTIFY_2000) and increases to 0.52 mW/m$^2$ in 2050 under the QUANTIFY_A1 scenario (**Table 4**). Based on earlier literature, Lee et al. (2009) reported a best estimate for this forcing of 2.8 mW/m$^2$ with a range of 0.39-20.3 mW/m$^2$. More recently Lee et al. (2021) reported values ranging from 0.4 mW/m$^2$ (Wilcox et al., 2012) to 1.5 mW/m$^2$ (Fromming et al., 2012; Lim et al., 2015) and even to 3.0 mW/m$^2$ (Penner et al., 2009), and derived a best estimate for this forcing of 2.0 (0.8, 3.2) mW/m$^2$. The forcing recalculated with the LMDZ-INCA chemistry-transport model is at the lower range of this previous literature.

Myhre et al. (2009) performed an intercomparaison of stratospheric water vapour radiative forcings in several models imposing idealized stratospheric $H_2O$ perturbations. In particular a simulation imposing an increase of water vapour in the stratosphere from 3 to 3.7 ppmv has been performed. In this case, based on 6 different models the calculated net radiative forcings at the tropopause range from 0.16 to 0.38 W/m$^2$ with a mean at 0.25 W/m$^2$. The longwave forcings range from 0.19 to 0.40 W/m$^2$ and the shotwave forcings range from -0.020 to -0.058 W/m$^2$. In order to evaluate our radiative transfer model, we have performed this benchmark simulation with the LMDZ-INCA model version used in the present study. We calculate a net radiative forcing at the tropopause of 0.18 W/m$^2$ (longwave: 0.22 W/m$^2$; shortwave: -0.038 W/m$^2$). These forcings agree with the model ranges provided by Myhre et al. (2009) for both forcing components.

The water vapour forcing associated with aircraft emissions calculated in this study is based on a detailed calculation involving stratospheric chemistry and transport and includes only the water vapour change above the tropopause. This forcing is at the lower range of previous estimates. It should be noted that these previous estimates are based on different model set-up. For instance, Wilcox et al. (2012) used a Lagrangian model for transport imposing a water vapour lifetime and Froming et al. (2012) provided a water vapour forcing including both the $H_2O$ change in the stratosphere and upper troposphere. In Brasseur et al. (2016), the reported stratospheric water vapour forcings by two models (1.3 and 2.0 mW/m$^2$) did account for both direct $H_2O$ emissions and photochemical production by methane. A more detailed evaluation of this stratospheric water vapour forcing associated with direct aircraft emissions is still needed based on various models using the same protocol for calculation.

### 6.5 Total direct radiative forcing from atmospheric composition changes

The total radiative forcing from aircraft emissions and associated with changes in atmospheric chemistry and direct aerosols forcings are given in **Table 4**. The total forcing calculated with the LMDZ-INCA model are negative for both the "present-day" and future (2050) simulations. The total forcing ranges from -2.1 mW/m$^2$ for the "present-day" inventory REACT4C_2006 (-3.1 mW/m$^2$ in the case of the QUANTIFY_2000 inventory) to -2.5 mW/m$^2$ in 2050 for the QUANTIFY_B1_ACARE scenario and to -10.4 mW/m$^2$ for the QUANTIFY_A1 scenario. A total negative forcing from reactive species and aerosol direct forcings was also calculated by Unger et al. (2013) for both present and future conitions based on the AEDT emission inventories. The total forcing associated with $NO_x$ only emissions (REACT4C_2006_NOx_Only) also becomes negative (-0.93 mW/m$^2$) when the indirect forcings from sulfates and nitrates particles are considered. The comparison with the baseline simulation (REACT4C_2006) indicates that the climate effect of $NO_x$ emissions amounts to about 45% of the total perturbation. In the future simulations, this translates into a decrease of the total forcing by about 30% when the $NO_x$ emissions are reduced by 50%. Since sulfates dominate the particulate radiative forcing from aircraft,





lowering the fuel sulfur contain is another potential mitigation option to reduce the total forcing. For the future QUANTIFY_A1 simulation, the use of desulfurized fuel reduces the total forcing by more than 45%.

The net effect of decreasing the flight cruise altitude by 2000 ft is to increase the total negative forcing from -2 mW/m² to -3.2 mW/m² (+57%). Increasing the flight altitude by 2000 ft decreases the negative forcing to -0.7 mW/m² (-65%). The variation of the total forcing with flight altitude is dominated by the high sensitivity of the ozone positive forcing to the altitude of the perturbation, with the variation of the negative sulfate forcing of secondary importance for these sensitivity simulations.

The radiative forcings from aircraft emissions calculated in this study are compared to the recent review by Lee et al. (2021) at **Table 6**. For this comparison the forcings are scaled to the 2018 aircraft fuel and the ERF/RF ratios provided by Lee et al. (2021) are applied. For the methane forcing, we also apply the transient correction factor (see **Section 6.2**) recalculated and applied by Lee et al. (2021) for the best estimates (0.79). Overall the agreement for the total forcing between this study and the best estimate from Lee et al. (2021) and based on the previous 1060 literature is reasonable. However, a detailed comparison shows that both the ozone positive and methane negative forcings are lower in LMDZ-INCA compared to the Lee et al. (2021) assessment. The ozone forcing calculed in this study is in better agreement with the Holmes et al. (2011) best estimate as discussed earlier (see also **Table S4**). As indicated above the stratospheric water vapour forcing is also significantly lower and requires further attention. It is also important to note that the total ERF is now positive and equal to 8.9 mW/m² while the total 1065 radiative forcing calculated in **Table 4** was negative (-2.06 mW/m²). This is due to the increase, in Lee et al. (2021), of the ozone ERF by almost 40% compared to the radiative forcing while keeping the aerosol forcings unchanged (ERF/RF = 1). Considering the uncertainty on the determination of the ERF/RF ratio for the different forcing agents, which depends on the climate model used, this feature also clearly requires further investigation. A similar conclusion applies to the methane transient factor estimate.

**7 Summary and conclusion**

Aviation $NO_x$ emissions have not only an impact on global climate by changing ozone and methane levels in the atmosphere but also contribute to deteriorate local air quality. Improved combustor performance from future 1075 aircraft can therefore contribute to reduce the impact of aircraft on climate but also provide a co-beneficial improvement of air quality. However, historically, reductions in aircraft $NO_x$ emissions have tended to increase fuel burn and hence $CO_2$ emissions. Thus, a trade-off generally arises between reducing aircraft $NO_x$ and $CO_2$ emissions. In order to properly assess the co-benefit with air quality improvement and the trade-off with climate change due to $CO_2$ emissions, it appears essential to better quantify the climate impact of aircraft $NO_x$ emissions 1080 and its future evolution.

In this paper, a new version of the LMDZ-INCA global model is applied to reevaluate the impact of aircraft $NO_x$ and aerosol emissions on climate. This version of the model is better designed to investigate the role played by aircraft emissions in the upper-troposphere and lower-stratosphere and includes both the gas phase chemistry in 1085 the troposphere and in the stratosphere. The model results have been compared to ozone soundings and to IAGOS data for $O_3$ and CO in the upper-troposphere and lower-stratosphere. In addition, tropospheric aerosols are also considered including the sulfate-nitrate-ammonium cycle and heterogeneous reactions between gas phase chemistry and aerosols. With this model, we investigate the impact of "present-day" baseline and future (2050) aircraft emissions on atmospheric composition and the associated radiative forcings of climate of ozone, methane 1090 and the aerosol direct forcings.

A major uncertainty regarding the future impact of aircraft $NO_x$ emissions arises from the scenarios that are used for the evolution of air traffic, aircraft operations, fuel used and assumptions on emission indices. In this study, we use a set of scenarios prepared in the framework of the previous projects QUANTIFY and REACT4C as they 1095 can be considered as benchmark scenarios, used in numerous former model simulations and intercomparisons. They are augmented by several scenarios designed to investigate selected mitigation options. This set of scenarios is generally consistent with more recent aircraft scenarios in terms of global mean emissions and provide a reasonable estimate for both baseline and mitigation scenarios.

The results from the various simulations performed in this study confirm that the efficiency of $NO_x$ to produce ozone is very much dependent on the injection height. For the baseline "present-day" REACT4C_2006 simulation, this efficiency is 6.6 $TgO_3$/TgN compared to an averaged efficiency of 1.1 $TgO_3$/TgN derived for instance by Hoor et al. (2009) for road transport emissions. The ozone response to aircraft emissions is therefore very much dependent on the cruise altitude and confirms the need for a fine vertical resolution in the models in the region of 1105 the upper-troposphere and higher-stratosphere in order to represent more accurately the aircraft emission vertical distribution. For instance, in the case of the QUANTIFY_2000 baseline "present-day" emissions this ozone production efficiency is slightly reduced to 6.0 $TgO_3$/TgN as a consequence of emissions injected at slightly lower cruise altitudes than REACT4C_2006 (Skowron et al., 2013). It implies that flying higher increases the impact of $NO_x$ on ozone (efficiency increased to 7.5 $TgO_3$/TgN when the altitude is increased by 2000 ft, 610 m) and flying 1110 lower reduces the impact on ozone (efficiency reduced to 5.8 $TgO_3$/TgN when the altitude is decreased by 2000 ft). In the future, an ozone production four times higher than for the "present-day" and reaching more than 20



TgO$_3$/TgN is calculated for the QUANTIFY_A1 scenario. An increase which is roughly linear with the increase in NO$_x$ emissions. The future evolution of the ozone production efficiency indicates that: 1) The efficiency increases with the background methane and NO$_x$ concentrations. In the case of the QUANTIFY_B1 and QUANTIFY_B1_ACARE scenarios, the efficiency increases to 7.7 and 7.9 TgO$_3$/TgN, respectively, for aircraft NO$_x$ emissions close to the baseline "present-day" values; 2) The efficiency increases with decreasing aircraft NO$_x$ emissions: 6.1, 6.8, 7.7 and 7.9 TgO$_3$/TgN for the A1, A1_LowNOx, B1 and B1_ACARE, respectively. These variations in ozone efficiency translate to the ozone radiative forcings as they exhibit, for both "present-day" and future scenarios, a fairly constant value per ozone change of 3.4 mW/m$^2$/TgO$_3$ (37.7 mW/m$^2$/DU).

As a result of NO$_x$ aircraft emissions, the atmospheric methane sink is decreased by about 1% for the "present-day" simulations and by up to 4.7% in 2050 for the QUANTIFY_A1 scenario. We found the methane lifetime variation is less sensitive to the aircraft NO$_x$ emission location than the ozone change. Similarly, the methane sink appears slightly less sensitive than the ozone production to the amount of aircraft NO$_x$ emitted. This agrees with Holmes et al. (2011) and Skowron et al. (2021) who found ozone more responsive to aircraft emissions than methane. The change in CH$_4$ mixing ratio itself represents 75% of the total methane forcing. The indirect changes through long-term tropospheric ozone, stratospheric water vapour and oxidation to CO$_2$ contribute for 19%, 4% and 1%, respectively. The net NO$_x$ forcing (O$_3$ + CH$_4$) is largely affected by the revised CH$_4$ radiative forcing formula provided by Etminan et al. (2016) which increases by 15% the total CH$_4$ negative forcing. As a consequence, the ozone positive forcing and the methane negative forcing largely offset each other resulting in a slightly positive forcing for "present-day" simulations (1-3 mW/m$^2$). In 2050, the net forcing even turns negative (-0.28 mW/m$^2$ for the QUANTIFY_A1 scenario) due essentially to higher methane background concentrations. Since the methane sink appears slightly less sensitive than the ozone production to the amount of aircraft NO$_x$ emitted, the total NO$_x$ forcing switches back to a slight positive value for the mitigation scenarios B1 and B1_ACARE. Skowron et al. (2021) also reached the similar conclusion that the revision of the methane radiative forcing significantly reduced the aircraft NO$_x$ forcing and that, in the future, this forcing could even turn negative, providing a different perspective on the aircraft NO$_x$ emissions impact on climate.

Additional radiative forcings involving particle formation arise from aircraft NO$_x$ emissions. The increased OH concentrations associated with NO$_x$ emissions are responsible for an enhanced conversion of SO$_2$ to sulfate particles and an associated negative forcing. Similarly, NO$_x$ emissions are expected to increase the formation of nitrate particles, also responsible for an additional negative forcing. The sulfate-nitrate-ammonium cycle is however important to consider since the increased NO$_x$ emissions from aircraft indeed increase nitrates particles in the lower troposphere where ammonia concentration are higher. However, in the upper troposphere, increased sulfate concentrations favor the titration of ammonia to form ammonium sulfates leading to lower ammonium nitrate concentrations (despite the increased NO$_x$ and HNO$_3$ concentrations). Overall, the indirect forcings of sulfate and nitrate aerosols associated with NO$_x$ emissions is negative and is estimated to -3.0 mW/m$^2$/TgN for the baseline "present-day" simulation. When these aerosol radiative forcings are considered, the net NO$_x$ forcing, due to O$_3$ and CH$_4$, and estimated to +2 mW/m$^2$/TgN for the baseline simulation, turns from a positive value to a negative value even for present-day conditions. These indirect forcings need further investigation and are sensitive to the model sulfate-ammonium-nitrate scheme (Unger et al., 2011; Righi et al., 2013; Pitari et al., 2016; Brasseur et al., 2016). However, they suggest that, in addition to the increased methane negative forcing discussed above, indirect forcing from sulfate and nitrate particles could turn the aircraft NO$_x$ radiative forcing from positive to negative even for present-day conditions and not only under future scenarios.

The concentrations of several types of particles are perturbed by aircraft emissions and hence provide direct radiative forcings of climate. Emissions of soot particles induce a positive radiative forcing of 0.46 mW/m$^2$ for the "present-day" baseline simulation increasing to 1.9 mW/m$^2$ in 2050. Sulfates particles are also significantly affected by the aviation SO$_2$ emissions but also by the increased conversion of SO$_2$ to sulfates as discussed above. The negative forcing associated with sulfates, -3.9 mW/m$^2$ for the "present-day" baseline simulation and increasing up to -13.7 mW/m$^2$ in 2050, largely dominates the effect of the other particles. The sulfate direct radiative forcing is estimated to be associated for about 50% to the direct SO$_2$ and SO$_4$ aircraft emissions and for about 50% to the increased conversion of SO$_2$ to SO$_4$ at higher OH concentrations, and hence related to the NO$_x$ emissions. Since the residence time of particles is longer at higher altitudes, the total aerosol direct forcing, dominated by the sulfates negative forcing, decreases by 12% at lower flight altitudes and increases by 13% at higher flight altitudes. For the future simulation, the use of desulfurized fuel reduces the sulfate direct forcing and also has an impact on nitrates particles. In total, the use of desulfurized fuel decreases the forcing of particles (makes it less negative) from -12.4 mW/m$^2$ to -5.6 mW/m$^2$ in 2050 under the QUANTIFY_A1 scenario. We found the ozone forcing only slightly affected by heterogeneous chemistry in the upper-troposphere.

The radiative forcing associated with direct aircraft emissions of water vapour in the stratosphere is 0.13-0.16 mW/m$^2$ for "present-day" conditions and increases up to 0.52 mW/m$^2$ in 2050 under the QUANTIFY_A1 scenario. The water vapour forcing associated with aircraft emissions calculated in this study is based on a detailed calculation involving stratospheric chemistry and transport and include only the water vapour change above the tropopause. This forcing is at the lower range of previous estimates. It should be noted that these previous estimates are based on very different model set-up. This forcing is likely to be small, however, a more detailed evaluation



of this stratospheric water vapour forcing associated with direct aircraft emissions is still needed based on several models using the same protocol for calculation.

The total radiative forcing from aircraft emissions associated with changes in atmospheric chemistry and direct aerosols forcings is negative for both the "present-day" and future (2050) simulations. The total forcing ranges from -2.1 mW/m$^2$ for the "present-day" inventory REACT4C_2006 (-3.1 mW/m$^2$ in the case of the QUANTIFY_2000 inventory) to -2.5 mW/m$^2$ in 2050 for the QUANTIFY_B1_ACARE scenario and to -10.4 mW/m$^2$ for the QUANTIFY_A1 scenario. NO$_x$ emissions cause about 45% of the total forcing for the "present-

day". Since sulfates dominate the aerosol direct radiative forcing from aircraft, lowering the fuel sulfur content is a potential mitigation option to reduce this total negative forcing. For the future QUANTIFY_A1 simulation, the use of desulfurized fuel reduces the total forcing by more than 45%. The net effect of decreasing the flight cruise altitude by 2000 ft is to increase the total negative forcing from -2 mW/m$^2$ to -3.2 mW/m$^2$ (+57%). Increasing the flight altitude by 2000 ft decreases the negative forcing to -0.7 mW/m$^2$ (-65%). The variation of the total radiative

forcing with flight altitude is dominated by the high sensitivity of the ozone positive forcing to the altitude of the emissions and hence the aviation-induced perturbation, with the variation of the negative sulfate forcing being of secondary importance.

        Overall the agreement for the non-CO$_2$ total forcing between this study and the best estimate from Lee et al. (2021)
is reasonable. However, a detailed comparison reveals several key differences and points to the key uncertainties associated with the NO$_x$ and aerosol direct forcings. Both the ozone positive and methane negative forcings are lower in LMDZ-INCA compared to the Lee et al. (2021) assessment. This is mostly due to the increase, in Lee et al. (2021), of the ozone ERF by almost 40% compared to the radiative forcing while keeping the aerosol forcings unchanged (ERF/RF = 1). Considering the uncertainty on the determination of the ERF/RF ratio for the different
forcing agents, which depends on the climate model used, this feature clearly requires further investigation. A similar conclusion applies to the methane transient factor estimate which can flip the sign of the total NO$_x$ forcing. The stratospheric water vapour forcing due to direct aircraft emissions is also significantly lower in this study. This forcing, although quite small for subsonic aircraft emissions, also requires further attention. The role played by nitrates particles has not been assessed in Lee et al. (2021). This indirect term associated with NO$_x$ emissions
is also a topic for further studies.

        Several previous studies have suggested that cruise emissions could be a significant or even dominant contribution to aviation-attributable surface level particulate matter and ozone (e.g., Barrett et al., 2010; Unger, 2011; Hauglustaine and Koffi, 2012; Lee et al., 2013; Eastham and Barrett, 2016). Our results confirm this important
subsidence of ozone produced in the free troposphere by cruise altitude NO$_x$ emissions, down to the lower troposphere and surface. A similar downward transport of black carbon and sulfates is also simulated by the global model with a significant increase of particulate matter levels at the surface. We further find that aircraft NO$_x$ emissions are responsible, in the lower troposphere, for an important formation of ammonium nitrate particles, in particular in regions of high ammonia concentrations. Since ammonia concentrations are predicted to increase in
the future (Hauglustaine et al., 2014), the aircraft NO$_x$ emissions could become a significant source of nitrate particles near the ground, in addition to contributing already significantly to ground-level sulfates and ozone concentrations.

        These results show that several mitigation options involving aircraft flight operation and cruise altitude changes,
traffic growth, engine technology, and fuel type, exist to reduce the climate impact of aircraft NO$_x$ emissions. However, our results, based on a current state-of-the-art global model, also show that the climate forcing of aircraft NO$_x$ emissions is likely to be small or even turn to negative (cooling) depending on atmospheric NO$_x$ or CH$_4$ future background concentrations and when the aircraft NO$_x$ impact on sulfate and nitrate particles is considered. There are still large uncertainties on the estimate of the aircraft NO$_x$ net impact on climate. In particular the use of
Effective Radiative Forcings (ERFs) or accounting for the methane transient forcing varies among the different models. These methodological concepts require further investigation in order to determine the best appropriate metric to express the climate impact of aircraft NO$_x$. Nevertheless, the results suggest that reducing aircraft NO$_x$ emissions is primarily beneficial for improving air quality and reducing O$_3$ and Particulate Matter (PM) ground-level concentrations. In line with the recent findings of Skowron et al. (2021), for climate consideration, one option
to reduce uncertainties in mitigation strategies might be to prioritize the reduction of CO$_2$ aircraft emissions which have a well-established and long-term impact on climate (Terrenoire et al., 2019), however this reduces the overall mitigation potential. New technologies in combustor design, which could help to reduce simultaneously NO$_x$ and CO$_2$ emissions, do not appear essential for aircraft non-CO$_2$ climate mitigation but rather as being co-beneficial for climate change mitigation from CO$_2$ emission reduction and for air quality improvement from reduced aviation
NO$_x$ emissions.

        ***Acknowledgements.*** This study was partly funded by the Direction Générale de l'Aviation Civile (DGAC) under the IMPACT and CLIMAVIATION projects and by the European Union Horizon 2020 research and innovation programme under the ACACIA project (Grant Agreement No 875036). The simulations were performed using
HPC resources from GENCI (Grand Equipement National de Calcul Intensif).





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






**Table 1.** Total global aircraft emissions corresponding to the baseline REACT4C_2006 and QUANTIFY_2000 inventories.

| Species | Units | REACT4C_2006 | QUANTIFY_2000 |
|---|---|---|---|
| Fuel | Tg yr$^{-1}$ | 178 | 214 |
| CO$_2$ | Tg yr$^{-1}$ | 560 | 672 |
| NO$_x$ | TgN yr$^{-1}$ | 0.71 | 0.84 |
| BC | Gg yr$^{-1}$ | 4.0 | 5.0 |
| SO$_2$ | GgS yr$^{-1}$ | 71 | 85 |
| HC | Gg yr$^{-1}$ | 63 | 91 |
| OC | Gg yr$^{-1}$ | 8.8 | 10.0 |
| SO$_4$ | GgS yr$^{-1}$ | 2.3 | 2.8 |
| H$_2$O | Tg yr$^{-1}$ | 220 | 264 |
| CO | Tg yr$^{-1}$ | 0.51 | 0.72 |


**Table 2**. Total global aircraft emissions for future 2050 baseline scenario QUANTIFY_A1 and for the two mitigation scenarios QUANTIFY_B1 and QUANTIFY_B1_ACARE.

| Species | Units | A1 | B1 | B1_ACARE |
|---|---|---|---|---|
| Fuel | Tg yr$^{-1}$ | 716 | 434 | 313 |
| CO$_2$ | Tg yr$^{-1}$ | 2257 | 1367 | 986 |
| NO$_x$ | TgN yr$^{-1}$ | 3.3 | 1.0 | 0.69 |
| BC | Gg yr$^{-1}$ | 16.0 | 8.9 | 6.4 |
| SO$_2$ | GgS yr$^{-1}$ | 280 | 170 | 120 |
| HC | Gg yr$^{-1}$ | 280 | 150 | 110 |
| OC | Gg yr$^{-1}$ | 35.0 | 21.0 | 15.0 |
| SO$_4$ | GgS yr$^{-1}$ | 9.3 | 5.5 | 4.1 |
| H$_2$O | Tg yr$^{-1}$ | 886 | 537 | 387 |
| CO | Tg yr$^{-1}$ | 2.3 | 1.3 | 0.93 |







**Table 3.** Annual aircraft $NO_x$ emissions ($ENO_x$, TgN), ozone burden variation ($\Delta O_3$, $TgO_3$), ozone to $NO_x$ sensitivity ($\Delta O_3/ENO_x$ , $TgO_3/TgN$), methane lifetime variation ($\Delta\tau_{CH4}$, %) and methane lifetime to $NO_x$ sensitivity ($\Delta\tau_{CH4}/ENO_x$ , %/TgN), for the various simulations.

| Scenario | $ENO_x$ | $\Delta O_3$ | $\Delta O_3/ENO_x$ | $\Delta\tau_{CH4}$ | $\Delta\tau_{CH4}/ENO_x$ |
|---|---|---|---|---|---|
| REACT4C_2006 | 0.71 | 4.67 | 6.58 | -1.01 | -1.43 |
| QUANTIFY_2000 | 0.84 | 5.05 | 6.01 | -1.15 | -1.37 |
| REACT4C_PLUS | 0.72 | 5.35 | 7.47 | -1.02 | -1.41 |
| REACT4C_MINUS | 0.71 | 4.15 | 5.84 | -1.01 | -1.42 |
| | | | | | |
| QUANTIFY_A1 | 3.31 | 20.33 | 6.14 | -4.67 | -1.41 |
| QUANTIFY_A1_LowNOx | 1.66 | 11.34 | 6.83 | -2.62 | -1.58 |
| QUANTIFY_A1_Desulfurized | 3.31 | 20.18 | 6.09 | -4.66 | -1.41 |
| QUANTIFY_B1 | 1.04 | 7.95 | 7.67 | -1.76 | -1.69 |
| QUANTIFY_B1_ACARE | 0.69 | 5.46 | 7.91 | -1.19 | -1.73 |



**Table 4.** Radiative forcings of ozone, methane, black carbon (BC), sulfates, nitrates, organic carbon (OC) and stratospheric water vapour calculated for the different simulations ($mW/m^2$). The radiative forcing of methane is the sum of 4 terms as depicted in **Table 5**.


| Scenario | $F_{O3}$ | $F_{CH4}$ | $F_{BC}$ | $F_{SO4}$ | $F_{NO3}$ | $F_{OC}$ | $F_{H2O}$ | **Total** |
|---|---|---|---|---|---|---|---|---|
| REACT4C_2006 | 15.87 | -14.69 | 0.46 | -3.87 | 0.07 | -0.03 | 0.13 | **-2.06** |
| REACT4C_2006_NOx_Only | 15.90 | -14.72 | 0.00 | -2.00 | -0.12 | 0.00 | 0.00 | **-0.93** |
| QUANTIFY_2000 | 17.19 | -16.69 | 0.54 | -4.33 | 0.06 | -0.04 | 0.16 | **-3.12** |
| REACT4C_PLUS | 17.72 | -14.80 | 0.49 | -4.38 | 0.12 | -0.04 | 0.17 | **-0.72** |
| REACT4C_MINUS | 14.28 | -14.68 | 0.44 | -3.40 | 0.03 | -0.03 | 0.10 | **-3.24** |
| | | | | | | | | |
| QUANTIFY_A1 | 70.56 | -70.84 | 1.88 | -13.66 | 1.29 | -0.16 | 0.52 | **-10.40** |
| QUANTIFY_A1_LowNOx | 39.29 | -39.45 | 1.86 | -10.88 | 1.69 | -0.15 | 0.52 | **-7.11** |
| QUANTIFY_A1_Desulfurized | 70.08 | -70.65 | 1.89 | -6.47 | -0.84 | -0.16 | 0.52 | **-5.63** |
| QUANTIFY_B1 | 27.57 | -26.43 | 1.12 | -7.38 | 1.17 | -0.10 | 0.32 | **-3.73** |
| QUANTIFY_B1_ACARE | 18.74 | -17.90 | 0.84 | -5.20 | 0.85 | -0.07 | 0.23 | **-2.52** |




**Table 5.** Decomposition of the total methane forcing $F_{CH4}$ into its various direct and indirect forcings. The forcings are associated with changes in methane lifetime (CH4-OH), tropospheric ozone production (CH4-O3), stratospheric water vapour production (CH4-SWV) and $CO_2$ production (CH4-CO2) (mW/m$^2$).

| Scenario | $F_{CH4-OH}$ | $F_{CH4-O3}$ | $F_{CH4-SWV}$ | $F_{CH4-CO2}$ | $F_{CH4}$ |
|---|---|---|---|---|---|
| REACT4C_2006 | -11.02 | -2.82 | -0.65 | -0.20 | **-14.69** |
| REACT4C_2006_NOx_Only | -11.04 | -2.83 | -0.65 | -0.20 | **-14.72** |
| QUANTIFY_2000 | -12.52 | -3.20 | -0.74 | -0.23 | **-16.69** |
| | | | | | |
| QUANTIFY_A1 | -52.85 | -13.96 | -3.22 | -0.81 | **-70.84** |
| QUANTIFY_A1_LowNOx | -29.37 | -7.8 | -1.80 | -0.45 | **-39.45** |
| QUANTIFY_A1_Desulfurized | -52.71 | -13.93 | -3.21 | -0.80 | **-70.65** |
| QUANTIFY_B1 | -19.66 | -5.25 | -1.21 | -0.30 | **-26.43** |
| QUANTIFY_B1_ACARE | -13.31 | -3.56 | -0.82 | -0.21 | **-17.90** |



**Table 6.** Comparison of ozone, methane, black carbon (BC), sulfates, nitrates, organic carbon (OC) and stratospheric water vapour Effective Radiative Forcings (ERFs) calculated for the different simulations (mW/m$^2$) and compared with Lee et al (2021). For this comparison we rescale the REACT4C_2006 calculated forcings based on the total aviation fuel for 2018 and apply the ERF/RF factors provided by Lee et al. (2021). For methane, 1665 we also applied the correction factor (0.79) recalculated by Lee et al. (2021) for the year 2018 to account for the non-steady-state $CH_4$ responses to $NO_x$ emissions.

| | $F_{O3}$ | $F_{CH4}$ | $F_{BC}$ | $F_{SO4}$ | $F_{NO3}$ | $F_{OC}$ | $F_{H2O}$ | **Total** |
|---|---|---|---|---|---|---|---|---|
| Lee et al. (2021) | 49.3 | -35.0 | 0.94 | -7.4 | - | - | 2.0 | **9.8** |
| This work | 40.1 | -25.3 | 0.86 | -7.1 | 0.14 | -0.06 | 0.24 | **8.9** |

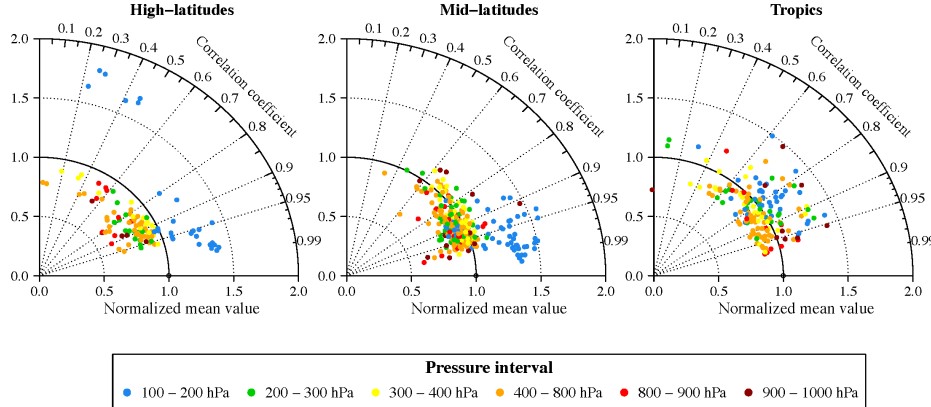

**Figure 1.** Taylor diagram comparing the mean and temporal correlation of ozone volume mixing ratio between ozonesonde climatology over the 1995-2011 period (Tilmes et al., 2012) and LMDZ-INCA model results, interpolated to the sample locations and interpolated on six pressure levels for high-latitudes, mid-latitudes and tropical stations.

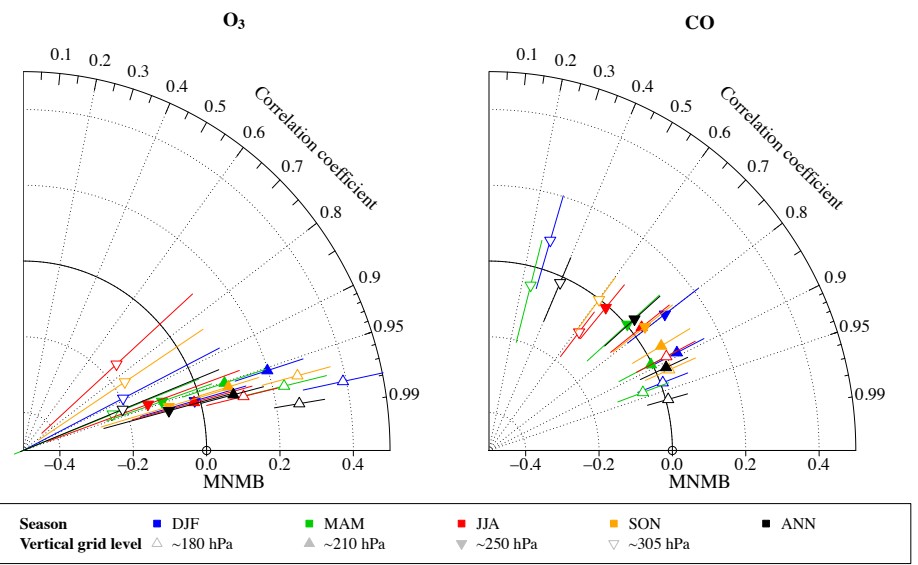

**Figure 2.** Taylor diagram comparing the mean and geographical correlation of ozone (left) and carbon monoxide (right) volume mixing ratio between the IAGOS data base (Cohen et al., 2018; 2021) and the LMDZ-INCA model results. The error bars denote the 1st and 3rd quartiles of the model to IAGOS biases for a given vertical level and season.



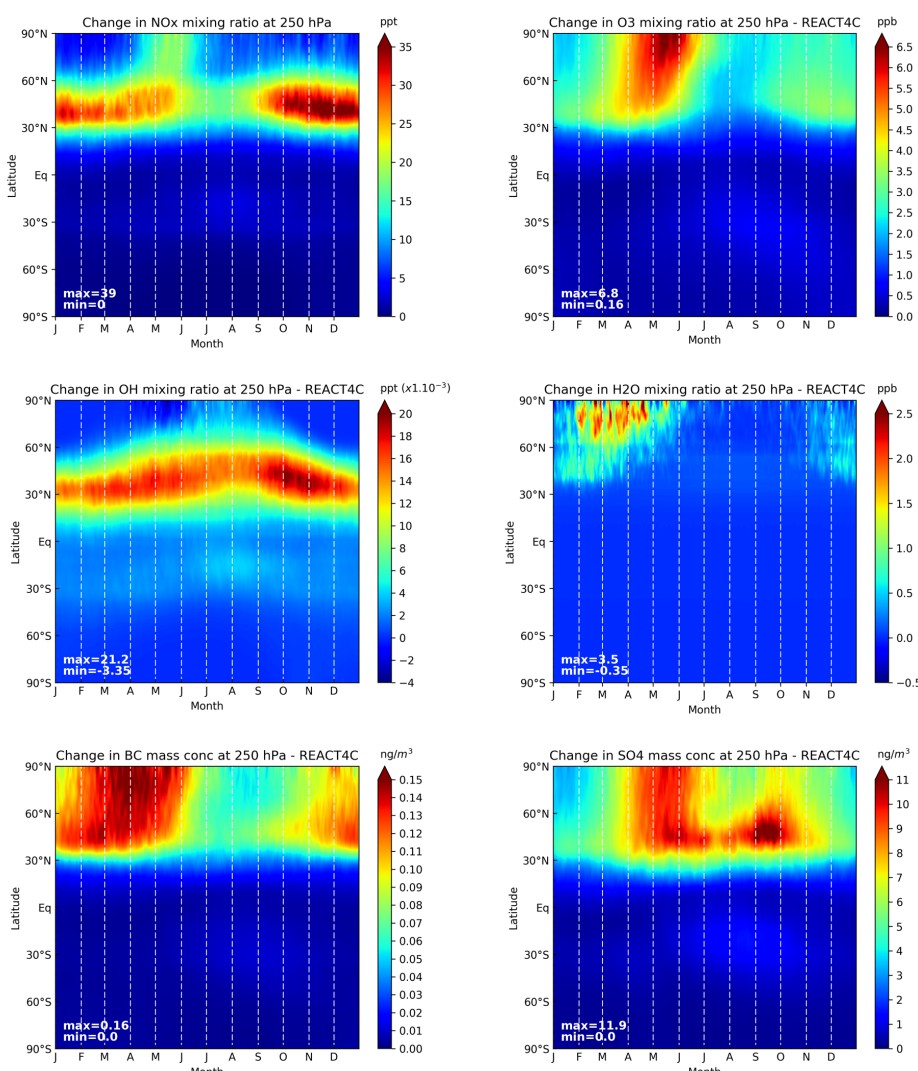

**Figure 3.** Daily and zonally averaged perturbation due to aircraft emissions at 250 hPa of $NO_x$ (pptv), $O_3$ (ppbv), OH ($10^{-3}$ pptv), $H_2O$ (ppbv), BC (ng/m³) and $SO_4$ (ng/m³) for the REACT4C_2006 inventory.



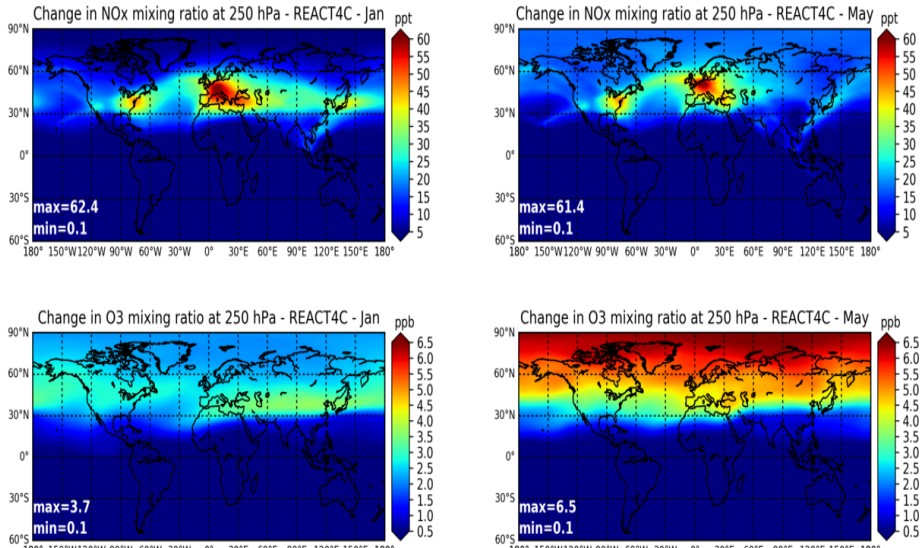

**Figure 4.** Spatial distribution of the 250 hPa perturbation due to aircraft emissions for the month of January (left) and May (right) for NOx (pptv) and ozone (ppbv) mixing ratio for the REACT4C_2006 inventory.

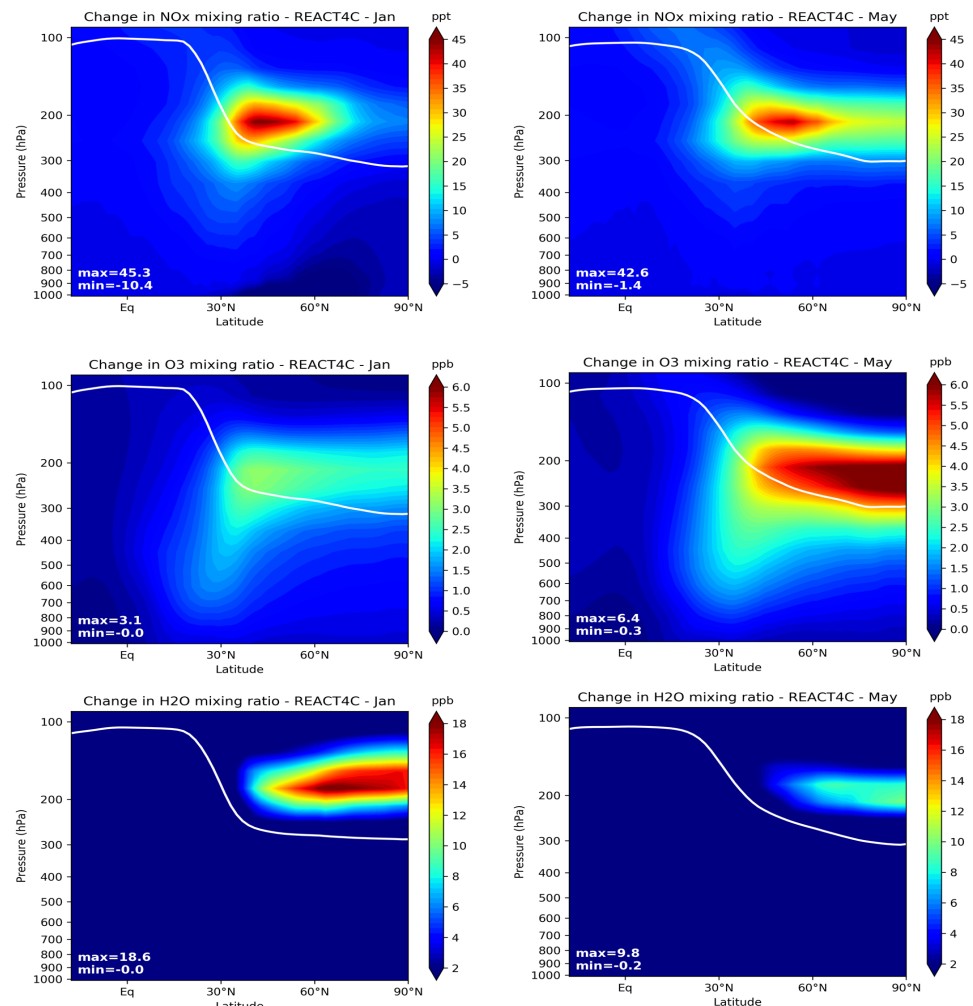

**Figure 5.** Zonal mean perturbation due to aircraft emissions for January (left) and May (right) of $NO_x$ (pptv), $O_3$ (ppbv), and $H_2O$ (ppbv) for the REACT4C_2006 inventory. The solid line represents the model tropopause pressure.

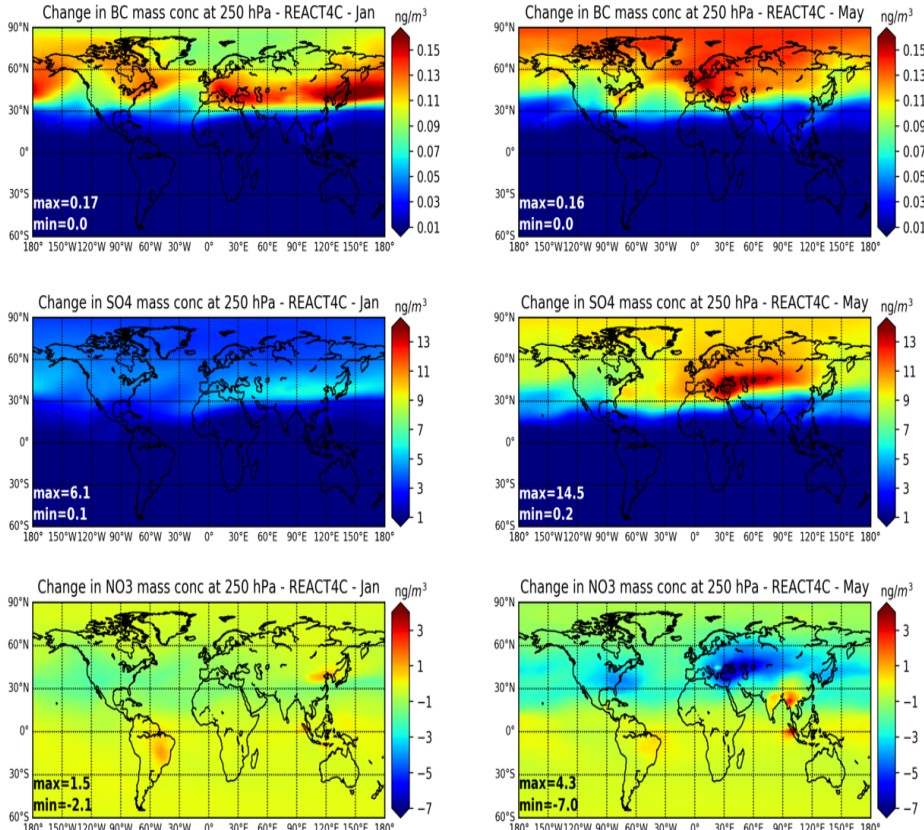

**Figure 6.** Spatial distributions of the 250 hPa perturbation due to aircraft emissions for January (left) and May (right) for BC, $SO_4$ and $NO_3$ (ng/m³) for the REACT4C_2006 inventory.

**Figure 7.** Zonal mean perturbation due to aircraft emissions for January (left) and May (right) of BC (ng/m$^3$), SO$_2$ (pptv), SO$_4$ (ng/m$^3$) and NO$_3$ (ng/m$^3$) for the REACT4C_2006 inventory. The solid line represents the tropopause pressure.



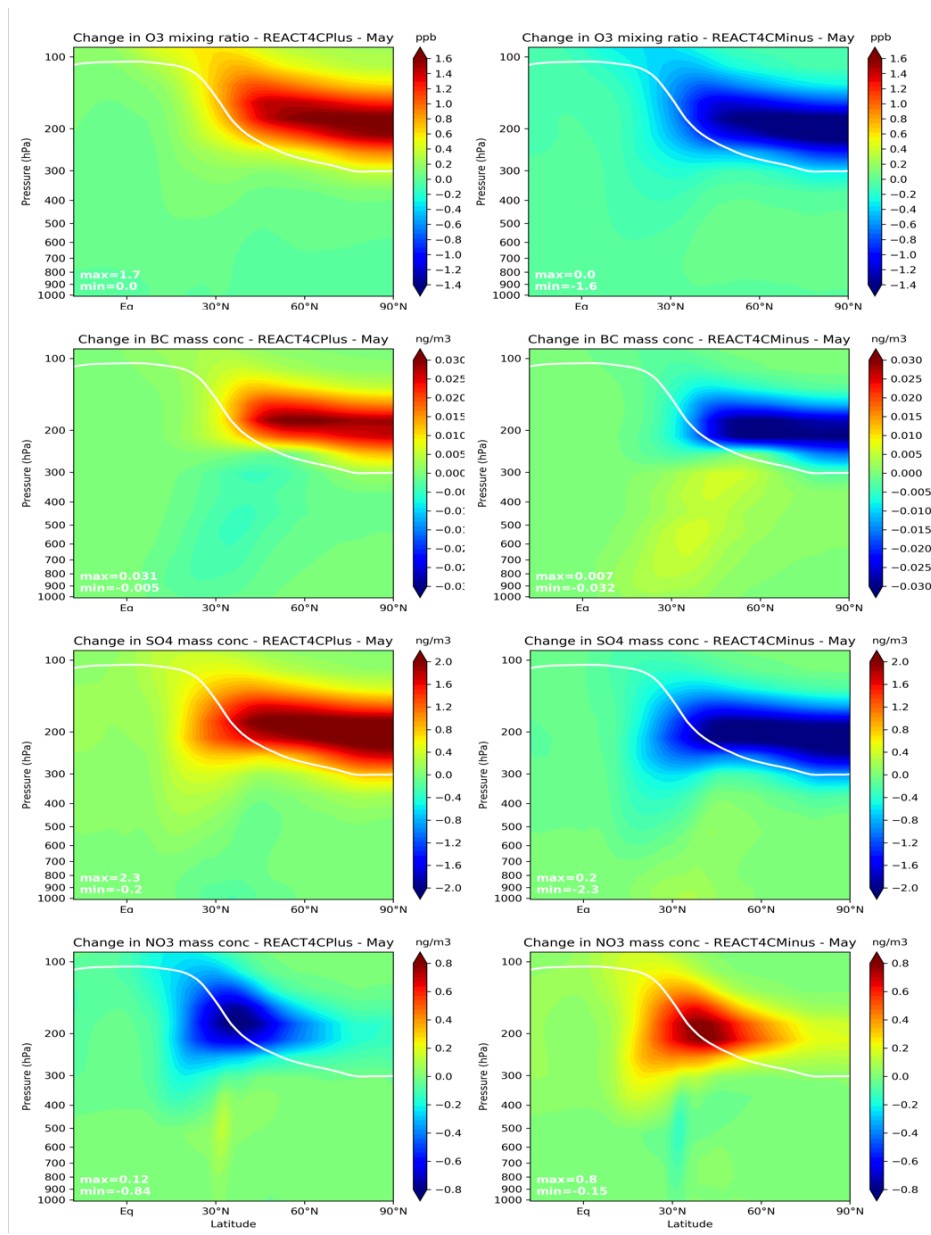

**Figure 8.** Zonal mean difference for the months of May for O$_3$ (ppbv), BC (ng/m3), SO$_4$ (ng/m$^3$) and NO$_3$ (ng/m$^3$) between the REACT4C_PLUS (left) and REACT4C_MINUS (right) simulations and REACT4C_2004 as the reference. The solid line represents the tropopause pressure.



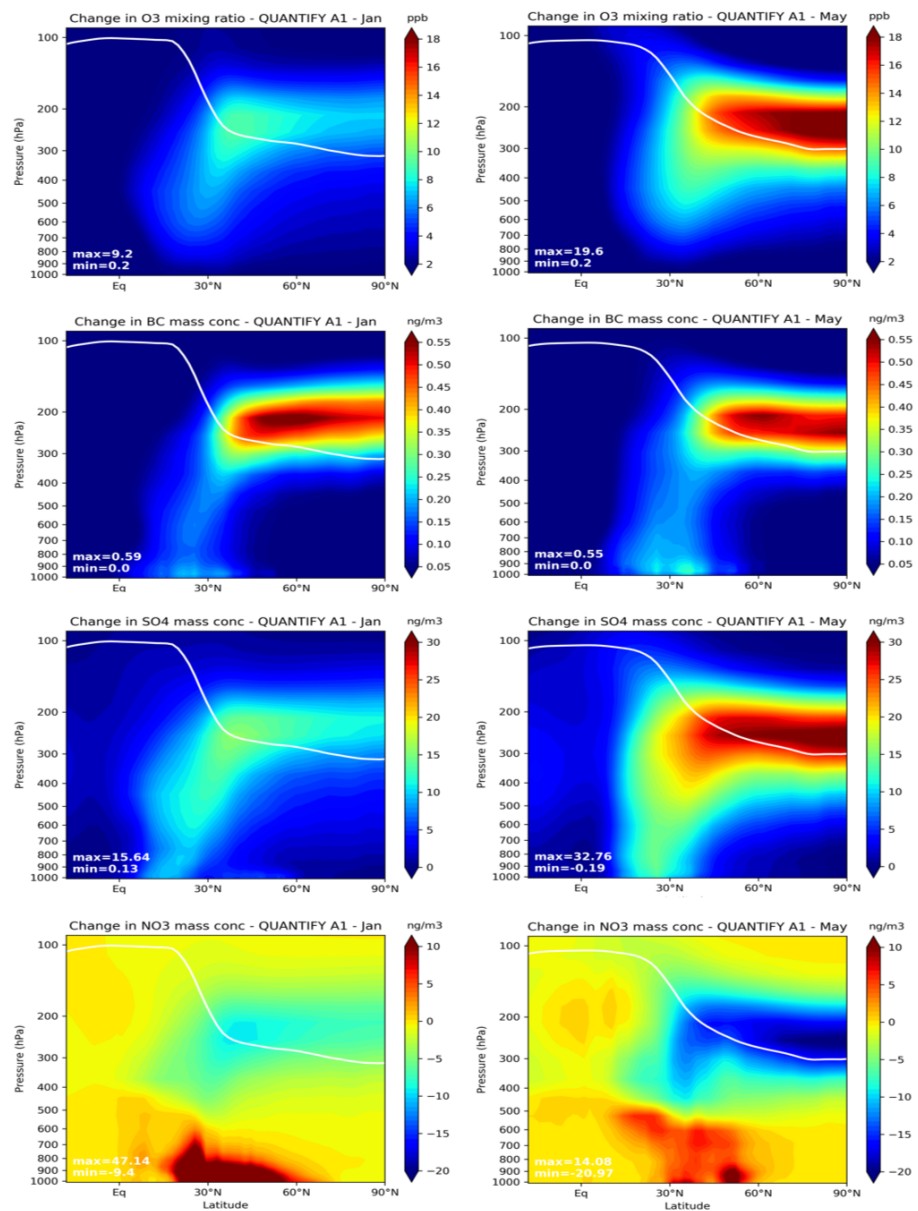

**Figure 9.** Zonal mean perturbation due to aircraft emissions for the months of January (left) and May (right) for $O_3$ (ppbv), BC (ng/m3), $SO_4$ (ng/m3) and $NO_3$ (ng/m3) averaged for the future scenario QUANTIFY A1 2050. The solid line represents the tropopause pressure.

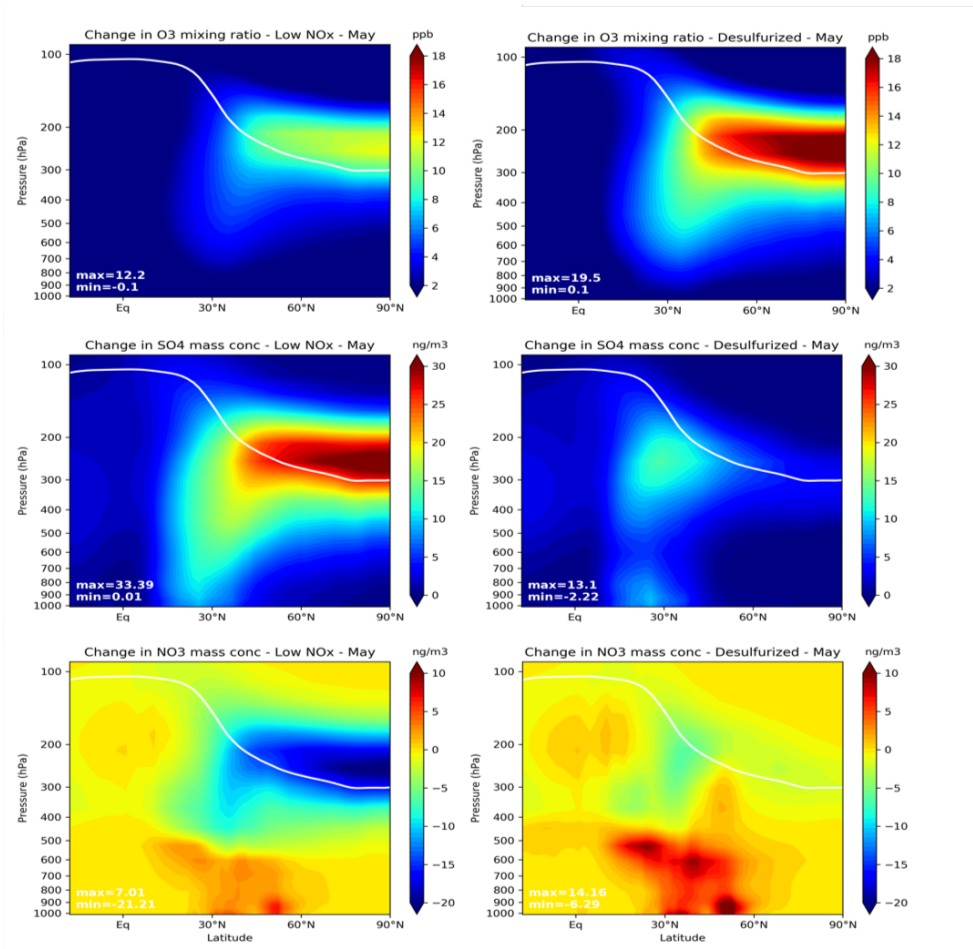

**Figure 10.** Zonal mean perturbation due to aircraft emissions for May of $O_3$ (ppbv), $SO_4$ and $NO_3$ (ng/m3) for the LowNOx (left) and Desulfurized (right) scenarios. The solid line represents the tropopause pressure.

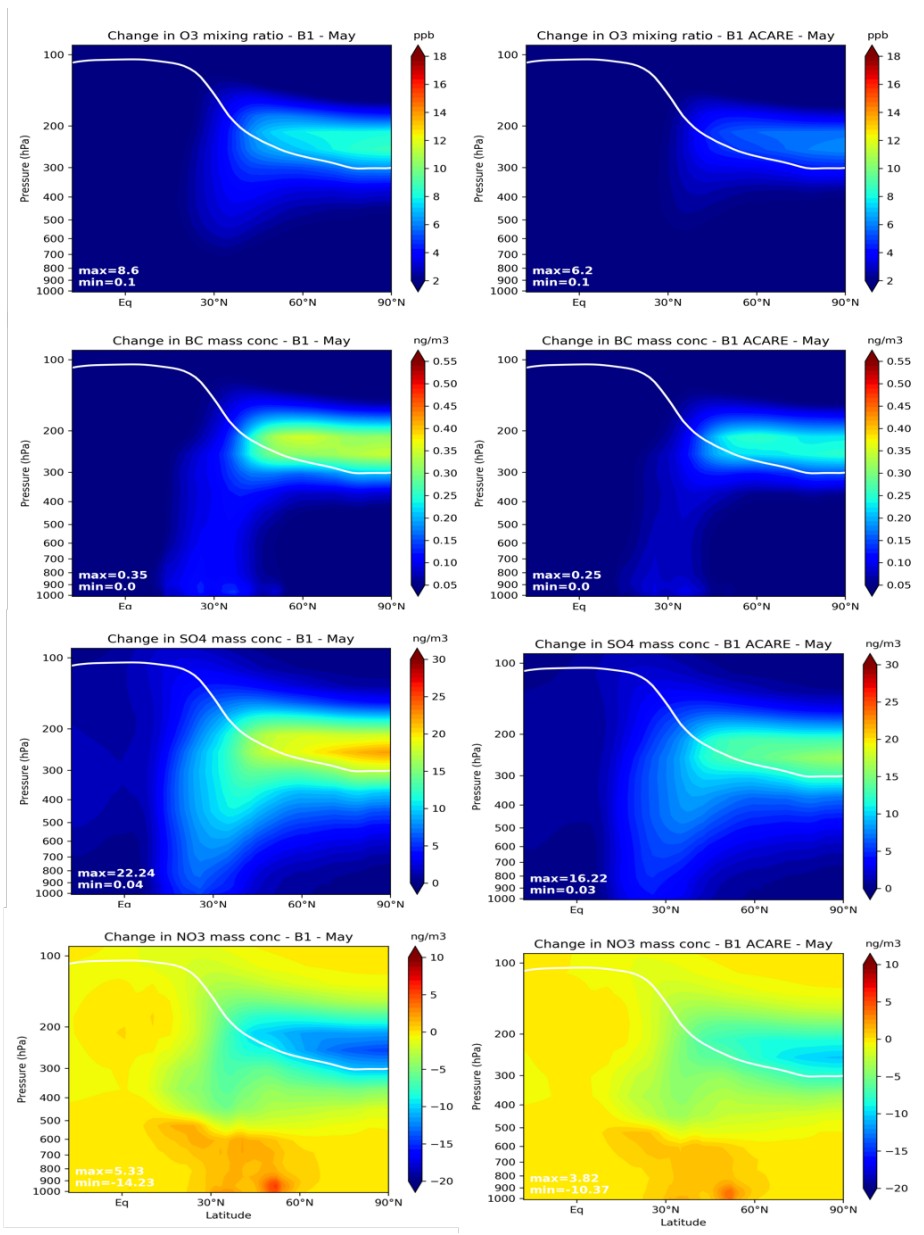

**Figure 11.** Zonal mean perturbation due to aircraft emissions for May of $O_3$ (ppbv), BC, $SO_4$, and $NO_3$ (ng/m3) for the QUANTIFY 2050 B1 (left) and B1 ACARE (right) scenarios. The solid line represents the tropopause pressure.