# Peer review of "Impact of present and future aircraft NOx and aerosol emissions on atmospheric composition and associated direct radiative forcing of climate"

_Atmospheric Chemistry and Physics, 2022_

## Author Comment (AC1)

**Author's replies to the reviewers**

**Impact of aircraft NOx and aerosol emissions on atmospheric composition and associated direct radiative forcing of climate**

**Terrenoire et al.**

We thank both reviewers for their comments on our manuscript and for the time spent on their reviews. Please find below a detailed point-by-point reply to the comments and suggestions to reviewer #1 and reviewer #2.

**Review 1 of "Impacts of present and future aircraft NOx and aerosol emissions on atmospheric concentrations and associated radiative forcing of climate" by Terrenoire et al.**

This paper studies the impact of present-day and a range of projected aviation NOx and aerosol emissions on atmospheric composition and climate. Aviation NOx has been the focus of numerous studies over the past decades, but here the authors include less well studied contributions from nitrate and sulfate and applies an updated and more refined, including the addition of interactive stratospheric chemistry, version of the LMDZ-INCA global model than previously used.

The results reconfirm previous findings and are in overall agreement with older comparable studies when it comes to features of the NOx-induced perturbations to ozone and methane concentrations. A key new finding is that accounting for indirect effects on nitrate and sulfate aerosols, results in a switch from net positive to net negative aviation NOx forcing. The sign of the net NOx forcing is also found to be highly sensitive to background conditions, with implications for future scenarios. Given that the most recent comprehensive assessment of aviation climate impact placed NOx as the third largest warming contribution, these are important findings. The sensitivity of the net NOx climate effect to these factors, combined with the large uncertainties surrounding atmospheric aerosol in general, shows that further efforts are required from the scientific community.

The paper is very well written and organized, albeit a bit long and detailed at times. The future scenarios used are old and may not capture more recent formulations of the ICAO goals and the Paris Agreement, which is an issue for the relevance of the results in the broader context – however, this is also understandable given the lack of up-to-date aviation emission scenarios and fine within the more process-focused scope of this paper.

Overall, I consider the paper to be an important contribution to the literature and have mostly minor comments for clarity and readability that should be considered before publication.

Thank you for the positive feedback. The abstract and the conclusion have been shortened and reworked (see also specific replies below). The future emission scenarios used in this paper are indeed somewhat dated. However, these scenarios can also be considered as reference scenarios with well-described hypothesis. We have tried to put them in the perspective of the newer scenarios comparing the total emissions to more recent scenarios like SSP or ACCRI scenarios. Also, various mitigation and alternate scenarios have been used in our paper in order to investigate newer hypothesis (reduced NO$_x$, desulfurized fuel). Interestingly, we found the QUANTIFY future emission scenarios generally in line with more recent scenarios as described

in the text (end of section 2.2) and Table S2. Please note that simulations based on the SSP scenarios are currently underway with this model and other CTMs in the framework of the ACACIA EU project. The results are unfortunately not yet available but will be documented in forthcoming publications.

General comment:

Both the abstract and Summary and conclusion sections are quite long and detailed, which makes it a bit challenging to extract the key findings and implications. I would encourage the authors to look for possibilities to reduce the level of detail in the abstract, which reads almost like a full summary on its own, as well as eliminate some of repetition that seems to be between the summary and previous sections (or alternatively remove from results section for a more condensed summary of course).

The abstract and the conclusion sections have been revised. The abstract is now shorter by about 50%. The conclusion section is also shorter by 1/3 and more focused on our key findings that we now enumerate in order not to repeat the previous sections.

The study includes a large number of experiments; while described in the methods section, I think it could be helpful for the reader to have a table that summarizes them.

A new Table 3 has been added in our manuscript, providing the labelling and key information on the various model simulations performed in the study.

Minor comments:

Line 130: Lund et al. 2017 https://doi.org/10.5194/esd-8-547-2017 also included nitrate aerosols

We were not aware of this publication. The reference has now been added.

Line 147: missing an "us"?

Corrected.

Line 215: Because the paper repeats the importance of the emission altitude and to be able to better study the zonal concentration changes, it would be useful with the figure (in the SI) showing the zonal mean, vertical emission distribution.

We have now added a new Fig. S1 showing the zonal mean distribution of the REACT4C and QUANTIFY fuel, $NO_x$ and BC aircraft emissions.

Lines 228-235: could be worth a quick mention of the evolution of the sector in the past 10-12 years (as assessed by Lee et al. 2021) since these emission inventories are older – for context. Also how, do the CO2 emissions for 2006 compare with those in Lee et al.? I seem to recall that CEDS emissions are a bit lower than that study in 2018, also the case for 2006?

We have added a short paragraph comparing the emissions as summarized in Table 1 and Table S1 to the 2000, 2006 and 2018 emissions calculated by Lee et al. (2021) and used also to rescale the forcings given in Table 7.

*The QUANTIFY_2000 $CO_2$ emissions are close to the estimate of 686 $TgCO_2$/yr for 2000 provided by Lee et al. (2021). In 2006, Lee et al. (2021) estimated an emission of 745 $TgCO_2$/yr, larger than the REACT4C_2006, ACCRI and CEDS inventories by 33%, 25%, and 4%, respectively. Over the 2006-2018 period, Lee et al. (2021) estimated an increase of 39%, reaching 1034 Tg $CO_2$/yr in 2018. The radiative forcings calculated with the REACT4C_2006 inventory will be rescaled based on these $CO_2$ emissions for 2018 in order to be compared to the forcings provided by Lee et al. (2021) at the end of this study.*

Lines 294: the study by Ocko et al. https://acp.copernicus.org/articles/19/14949/2019/ presented some scenarios that where aligned with ICOA goals and policies – worth mentioning comparing here, or in the discussion?

We focused (in Table S2 in particular) on emission inventories (SSP, ACCRI) of chemicals provided as three-dimensional fields, needed in the global models. The reference to C. Ivanovich, I. Ocko et al. (2019) (based mostly on $CO_2$ projections and RCP scenarios for $NO_x$ and aerosols) has been added in the introduction together with the reference to Terrenoire et al. (2019) also discussing future aircraft $CO_2$ emissions to 2100.

Line 380: given that one of the main novel contributions in this work is the inclusion of aerosols, I suggest the authors also add a brief summary of what has been found regarding model performance for sulfate/nitrate, rather than just referring to other studies.

Figure S2 comparing the modeled AOD to the MODIS and AERONET measurements has been added in the supplement. In addition, in the main paper, the following text has been added.

*The distribution of aerosol surface concentrations (sulfates, nitrates, ammonium) calculated with this version of the model has been evaluated and compared to surface stations by Hauglustaine et al. (2014) over Europe and Northern America and by Li et al. (2016) over China and is not repeated here. However, Fig. S2 compares the calculated total Aerosol Optical Depth (AOD) at 550 nm to the AERONET and MODIS data observations for Eastern and Central China, Western Europe and the Eastern United-States. A very good agreement is obtained with MODIS data with biases of -10%, 1.4% and -6.1 % for these three regions, respectively, and a correlation coefficient r of 0.59, 0.55 and 0.86. Larger biases are calculated against AERONET data, in particular over regions characterized by a high aerosol loading as China. Biases of respectively -39%, 10% and 24% are obtained for the three regions with correlation coefficients of 0.5, 0.45, and 0.78.*

Line 390: typo

Typo corrected. "Summarizes"

Line 390: can you please specify the latitude bands, unclear what high latitudes are

Figure 1 has been modified to provide the latitude bands and the main text has been modified accordingly.

Line 405: mix of southern and northern hemisphere stations as I understand it – are these three stations the only ones located in the southern hemisphere high lats?

We have clarified the main text and Figure 1 caption. In fact, the figure shows in different colors different altitude ranges. These altitude ranges include different levels as reported by Tilmes et al. (2012). In the case of the 100-200 hPa range, 3 levels are included (125 hPa, 150hPa, and 200 hPa). The 6 outliers in Fig. 1 (High-Latitudes) correspond to the 3 south polar stations at levels 125 hPa and 150 hPa.

Line 470: maybe mention findings about non-linearities in the scaled small perturbation vs 100% removal approach?

We have added a few sentences in Section 3.3 in order to explain more explicitly the differences obtained when applying small versus full perturbations in previous studies.

*Koffi et al. (2010) compared the two methodologies and found that the 100% perturbation of transport emissions induces a 6% higher transport-induced ozone burden perturbation in the case of aircraft emissions. As noted by Søvde et al. (2014) the small perturbation approach is more adapted when different transport modes, affected differently by non-linearities, are compared. The 100% perturbation gives the overall effect of aircraft, without considering compensating effects from other emission sectors due to chemical non-linearity (Koffi et al., 2010; Søvde et al. 2014; Grewe et al., 2019).*

Line 487: what about the approach to derive methane forcing? Describe here?

The method used to calculate the methane forcing has now been moved to Section 3.4.

Line 603: to get a better feel for the magnitude of the changes, it would be useful to express some of this in terms of percentage of the baseline concentration…

We have updated the text and for most of the zonal mean changes we now provide the absolute change and the relative change (in %) in the description of the figures.

Line 620-629: while the sulfate change is pronounced towards the high latitudes, the nitrate change is focused much further south – can the authors provide a brief explanation? Meteorological factors?

The change of nitrates linked to the changes in $SO_4$ appears only if $NH_3$ is present. At these altitudes, this occurs mostly at latitudes lower than 60°. The zonal mean $NH_3$ distributions presented for instance by Bian et al. (2017) in her Fig. 3c illustrate that $NH_3$ is present in the upper troposphere around 30-60°N in most models and not at higher latitudes. A sentence mentioning this feature has now been added in the manuscript.

Lin 764-782: what would be the potential implications of allowing climate to change, instead of fixing meteorological conditions? Of significance?

Indeed, in this study, we follow the methodology adopted in most of previous work and assume unchanged climate between present-day and future simulations. The aircraft perturbation is rather small and using a fixed meteorology allows to focus on the impact of aircraft emissions and compare the present-day and future impact of aircraft emissions. A few studies accounted for climate change in their transportation (aircraft) perturbation studies (Olivié et al., 2012; Huszar et al., 2013). However, the impact of future climate change on aircraft forcings is not isolated in these studies (and it is difficult to do so). However, there is clearly a potential impact

of future changes in upper tropospheric temperature, humidity and dynamics on aircraft perturbations but these effects need to be investigated more systematically in forthcoming studies. The importance of these changes at the 2050 time-horizon is probably less crucial than for a 2100 time-horizon, but this needs to be investigated. This limitation is now stressed in our conclusion (see below) and is mentioned as a perspective for future activities. In addition, this limitation is also briefly mentioned in the "Model set-up" section and in the paragraph referred to by the reviewer.

*Following previous work methodology, in order to better emphasize the impact of aircraft emissions on atmospheric composition, the impact of future climate change on aircraft perturbations is not considered in our simulations performed based on a present-day, unchanged, meteorology in 2050. The simulations performed by Olivié et al. (2012) and Huszar et al. (2013) did account for future climate change, however, the impact of future climate on atmospheric composition changes due to aircraft emissions is not isolated. Changes in upper-tropospheric temperature, humidity, and dynamics have the potential to affect the response of the atmosphere to aircraft emissions. These perturbations are likely to be more pronounced at a longer time-horizon than at the 2050 timeframe considered in our simulations, but this topic is clearly a subject to be investigated in forthcoming studies.*

Line 897-898: I feel that there is a bit of a mismatch between one of the novelties of this paper and the only two-line mention of this here. For instance, in light of the large uncertainties surrounding aerosols RF, how confident are you in the net response?

The last sentence of this paragraph was misplaced and has been moved 2 paragraphs below, as a full paragraph is already dedicated to these indirect forcings from sulfates and nitrates 2 paragraphs down. We have further stressed in this paragraph but also in the conclusion and in the abstract that the forcing of aerosols is subject to large uncertainties and requires further investigation with several models in order to reach a more consolidated view.

Line 910: is table S4 the correct reference here or am I misunderstanding the sentences?

Table S4 seems indeed the right table here and shows the various forcings associated with aircraft $NO_x$ emissions only and compares the ozone and various methane forcings to Lee et al. (2021) and Holmes et al. (2011). In addition, this table also provides the indirect forcings associated with sulfate and nitrate perturbations induced by the aircraft $NO_x$ emissions.

Line 1013: typo

Corrected

Line 1055: given the uncertainties surrounding the ERF/RF conversion factors, why not include a comparison with the RF numbers provided by Lee et all. (as well).

Table 7 and the corresponding text have been updated in order to include the RFs in the comparison. In addition, the ranges provided by Lee et al. (2021) are now also provided. Since the total forcings provided here are different from Lee et al. (2021) (not the same terms are included in this study), we recalculated the 5-95% percentiles based on the excel spreadsheet provided by these authors in order to use their discrete pdf distributions of the RFs and ERFs. A Monte-Carlo sampling similar to the one used by Lee et al. (2021) with 1 million sampling has been used to generate the total uncertainty.

---

## Author Comment (AC2)

**Author's replies to the reviewers**

**Impact of aircraft NOx and aerosol emissions on atmospheric composition and associated direct radiative forcing of climate**

**Terrenoire et al.**

We thank both reviewers for their comments on our manuscript and for the time spent on their reviews. Please find below a detailed point-by-point reply to the comments and suggestions to reviewer #1 and reviewer #2.

**Review 2 of the "Impact of present and future aircraft NOx and aerosol emissions on atmospheric composition and associated direct radiative forcing on climate" by Terrenoire et al.**

This study explores and gathers various aspects of the impact of aircraft NOx emissions on atmospheric composition and climate utilizing a series of present and future aviation projections, and an updated global chemistry-aerosol-climate model, LMDZ-INCA. In addition to the well-established effects of aircraft NOx emission, this paper also investigates the impacts that arise from the formation of nitrate and sulfate aerosols that constitute a novelty of the publication as it has been addressed in only a few modeling studies so far. It is also associated with large uncertainties, which might be worth highlighting more in this publication.

The paper is well-written and scientifically sound. The applied methods are valid and clear. The findings of the study are of interest to the community. I recommend this paper for publication, preceded by a few minor comments.

We thank the reviewer for the positive comments. We have revised the abstract and the conclusion and in particular we have tried to emphasize the uncertainties associated with the aerosol forcings in these sections.

General comments:

Abstract reads more like a summary. It is too long, with too many technicalities, and it is difficult to follow. I would suggest authors consider re-writing it, concentrating on the main findings and their implications.

The abstract has been revised and is now shorter by about 50%.

The text is full of details, which on one side is understandable taking into account the number of experiments that were performed for this study. On the other hand, maybe authors can help to level this complexity of the paper by preparing a kind of 'takeaway figure'. Figure that would summarize the main results, highlight the novelty of this study, etc. I believe the paper might gain the readability by having such a figure, but I leave this decision with the authors.

The conclusion section has been revised. It is now shorter by 1/3 and more focused on the take-away messages that we now enumerate in order not to repeat the previous sections and to provide the take away messages. Please note also that a new Table 3 has been added in our manuscript, providing the labelling and key information on the various model simulations performed in the study.

Specific comments:

Lines 90-94: some references can be useful here

Two references have been added in the manuscript on these issues: Freeman, S., D. S. Lee, L. L. Lim, A. Skowron, and R. Rodriguez De Leon, Trading off aircraft fuel burn and NOx emissions for optimal climate policy. *Environ. Sci. Technol.* 52, 2498−2505 (2018) and Prashanth, P., R. L. Speth, S. D. Eastham, J. S. Sabnis, and S.R.H. Barrett, Post-Combustion Emissions Control in Aero-Gas Turbine Engines. *Energy and Environmental Science*, DOI: 10.1039/D0EE02362K, 2021.

Line 219: motigtaion ---> mitigation

The typo has been corrected.

Line 239: the list (e.g., as a Table) of performed experiments, maybe with their short description, might be helpful

A new Table 3 has been added in our manuscript, providing the labelling and key information on the various model simulations performed in the study.

Lines 464: why your "present-day" is 2004 here, while in the previous section is 2006?

It is indeed 2006. The typo has been corrected.

Line 490: the description of methane RF calculation and all the methane-induced components are missing? Maybe part of your section 6.2 can be moved to 3.4, or vice versa, for consistency.

The method used to calculate the methane forcing has now been moved to Section 3.4.

Line 528: Figure 5 (and subsequent Figures) why May? Why not July, or JJA? On the other hand, your RF numbers are based on annual averages, so maybe your chemistry analysis can show the annual means too?

As shown in Figure 3 the maximum perturbation in ozone associated with aircraft emissions is calculated in May in this model. This figure was specifically introduced to illustrate the full seasonal cycle, before focusing on the minimum (January) and maximum perturbations (May). May was specifically chosen to illustrate this maximum. In contrast to some models showing a maximum in July or in summer, another category of models (including LMDZ-INCA) shows a maximum in late spring. This maximum reached in May at high latitudes is the result of a combination of a more intense photochemical activity in summer combined with a more intense transport poleward in spring. As a consequence, the peak is reached at the end of the spring season. We have added a sentence when describing Fig. 3 in order to better explain this and the choice of May is better justified when introducing Fig. 5. For the zonal mean perturbations we would like to keep this illustration of the minimum and the maximum. For the future mitigation scenarios we only illustrate the maximum in order to have a clear signal and not duplicate the number of figures, but the minimum perturbation is also provided in the text. This way, the seasonal cycle is illustrated in Fig. 3, the zonal perturbations show the min and max

perturbations, and the budgets and forcings provide the annual mean perturbations more important for climate purposes. We hope this provides a consistent set of illustrations.

Line 657: since you discuss the future scenarios run with interactive chemistry, it would be interesting to know how future climate change might influence your results. What is the sensitivity of aviation forcings to future climate? Is the use of present-day meteorology justified here?

Indeed, in this study, we follow the methodology adopted in most of previous work and assume unchanged meteorology between present-day and future simulations. The aircraft perturbation is rather small and using a fixed meteorology allows to focus on the impact of aircraft emissions and to compare the present-day and future impact of aircraft emissions. A few studies accounted for climate change in their transportation (aircraft) perturbation studies (Olivié et al., 2012; Huszar et al., 2013). However, the impact of future climate change on aircraft forcings is not isolated in these studies (and it is probably difficult to do so). There is clearly a potential impact of future changes in upper tropospheric temperature, humidity and dynamics on aircraft perturbations but these effects need to be investigated more systematically in forthcoming studies. The importance of these changes at the 2050 time-horizon is probably small but this needs to be investigated. This limitation is now stressed in our conclusion (see below) and is mentioned as a perspective for future activities. In addition, this limitation is also briefly mentioned in the "Model set-up" section and in the paragraph spotted here by the reviewer.

*Following previous work methodology, in order to better emphasize the impact of aircraft emissions on atmospheric composition, the impact of future climate change on aircraft perturbations is not considered in our simulations performed based on a present-day, unchanged, climate in 2050. The simulations performed by Olivié et al. (2012) and Huszar et al. (2013) did account for future climate change, however, the impact of future climate on atmospheric composition changes due to aircraft emissions is not isolated. Changes in upper-tropospheric temperature, humidity, and dynamics have the potential to affect the response of the atmosphere to aircraft emissions. These perturbations are likely to be more pronounced at a longer time-horizon than the 2050 timeframe considered in our simulations, but this topic is clearly a subject to be investigated in forthcoming studies.*

Line 682: wouldn't it be more suitable to discuss the effects of the mitigation scenarios based on the annual change, not a one-month response?

As mentioned above, for the zonal mean perturbations we would like to keep this illustration of the minimum and the maximum of the perturbations. For the future mitigation scenarios we indeed illustrate the maximum solely in order to have a clear signal and not duplicate the number of figures, however the minimum perturbations in January are also cited in the text. In this way, the seasonal cycle is illustrated in Fig. 3, the zonal perturbations show the min and max perturbations (or only the max with the min cited), and the budgets and forcings provide the annual mean perturbations more important for climate purposes. The choice of May is better justified when introducing Fig. 5.

Line 795: 7.2 ---> 7.3

We checked this value and confirm that Holmes et al. (2011) in their Table 2 do provide a decomposition factor of $21.6 \pm 7.2$ mW/m$^2$/TgN.

Line 833: having fixed surface CH4 concentrations means your CH4 and OH interactions are constrained, so it is not obvious how you derived your CH4 feedback factor.

This methodology based on fixed surface methane perturbations has been described in Prather et al. (2001) or in various IPCC reports and has been commonly used and applied in numerous studies including Holmes et al. (2011) and Fiore et al. (2009). We have applied this commonly used method in our study and derive a feedback factor $f$=1.36 in agreement with this previous work and the recent IPCC AR6. Other methodologies exist (Khodayari et al., 2015; Pitari et al., 2016) as mentioned in our manuscript and the fixed surface mixing ratio method was found accurate to within 10%. This sentence has been revised in our manuscript in order to reemphasize this limitation.

*Khodayari et al. (2015) concluded that for the simulations with fixed CH4 at the lower boundary condition, the parameterization based on Eq. (1) using the global mean lifetime approach overestimates the change in CH4 by 8.6% compared to the change calculated directly from the model using CH4 surface emissions. The overestimation is 12.1%-20.0% if using other lifetime approaches. They concluded that the parameterization based on Eq. (1) is good to within ~10% when using the global mean lifetime approach.*

Line 875 and Table 5: your long-term ozone estimate is much smaller than what can be found in other studies, e.g., 47% (Wild et al., 2001), 58% (Kohler et al., 2008), 42% (Hoor et al., 2009), 51%, 43% (Pitari et al., 2016) of the CH4 RF. Most of the latest aircraft studies include 50% (IPCC AR4, Myhre et al., 2013) in their calculations. The same applies to your stratospheric water vapor; most studies calculate it to be 15% (Myhre et al., 2007) of the CH4 RF, while yours is around three times smaller. These need some clarification or/and justification.

The methane indirect forcings that we calculate (paragraph now moved to section 3.4) are 116 mW/m$^2$/ppmvCH$_4$ for the long-term O$_3$ forcing and 27 mW/m$^2$/ppmvCH$_4$ for stratospheric H$_2$O. In the recent AR6 Chapter 7 (Section 7.6) these forcings are estimated as 140 ± 70 mW/m$^2$/ppmvCH$_4$ for long-term ozone and 40 ± 40 mW/m$^2$/ppmvCH$_4$ for stratospheric water vapour. This implies a forcing smaller by 17% for ozone compared to the IPCC best estimate but well within the provided confidence level. For stratospheric H$_2$O, the forcing is subject to a larger uncertainty, and we derive a forcing lower by 32% compared to the IPCC best estimate but also well within the (large) confidence level. The reference and the IPCC values now available have been added in our manuscript in Section 3.4.

Line 906: based the Etminan ---> based on the Etminan

Typo corrected.

Line 946: and 2004 ---> and 2005

Typo corrected.

Line 888 and Table S3: the CH4 increase via the Etminan parametrization you report is 15%, and it is smaller than what other aircraft studies calculate (e.g., Grewe et al., 2019), Skowron et al., 2021). Any explanation for that? At the same time, a few paragraphs above (line 822), you mention the 24.5% increase. It is confusing.

The methane forcing we calculate (without indirect effects except OH feedback) of -11.02 mW/m$^2$ (Table 6) is actually larger than Skowron et al. (2021) (i.e., -8.4 mW/m$^2$, their Table 2) reflecting the uncertainty on OH changes in our models (LMDZ-INCA in this study and MOZART3 in their study). The effect of the new Etminan parameterization is 22% in their case. It is more complicated to compare with Grewe et al. (2019) because they start from a methane forcing based on a different emission inventory (Lee et al., 2009) but they derive an increase of the forcing of 23% when the revised Etminan formula is used. The 15% increase provided in our study refers to Table S3 and to the total methane forcing (including the indirect ozone and water vapour effects). The impact of the revised Etminan formula on the methane forcing only (no indirect effects other that OH feedback) is also equal to 23% in our calculations. The 24.5% increase has been calculated by Etminan for a halving of the present-day methane concentration and is cited as an example. A reference to Etminan is provided to avoid confusion when the 24.5% value is cited and the 15% value for the total methane forcing is also better stressed. In addition, the 23% increase for the methane direct forcing only is now mentioned for clarity.

Line 1055: considering significant uncertainties associated with ERF/RF factors (especially those related to NOx as based on just one study according to Lee et al., 2021), wouldn't it be better to compare RF numbers? It would also be consistent with Holmes et al. (2011) comparison. Another aspect is the nature of these studies, Lee et al. (2021) and Holmes et al. (2011) are multi-model ensembles, so maybe presenting also ranges that they report together with their best estimates might help your comparison to be more feasible.

This is indeed an excellent idea in order to illustrate the large uncertainties on the forcings. Table 7 and the corresponding text have been updated in order to include the RFs in the comparison. In addition, the ranges provided by Lee et al. (2021) are now also provided. Since the total forcings provided here are different from Lee et al. (2021) (not the same terms are included in this study), we recalculated the 5-95% percentiles based on the excel spreadsheet provided by these authors in order to use their discrete pdf distributions of the RFs and ERFs. A Monte-Carlo sampling similar to the one used by Lee et al. (2021) with 1 million sampling has been used to generate the total uncertainty. Please note that in this Table 7 we only provide the values of Lee et al. (2021) often cited as a reference publication. The Holmes et al. (2011) results (only available for NO$_x$) are however used and compared to Lee et al. (2021) and to the calculated forcings in Table S4 dedicated to the NO$_x$ terms.